# Plant cysteine oxidases are dioxygenases that directly enable arginyl transferase-catalysed arginylation of N-end rule targets

Mark D. White[1], Maria Klecker[2,3], Richard J. Hopkinson[1], Daan A. Weits[4], Carolin Mueller[5,6], Christin Naumann[2,3], Rebecca O'Neill[1], James Wickens[1], Jiayu Yang[1], Jonathan C. Brooks-Bartlett[7], Elspeth F. Garman[7], Tom N. Grossmann[5,6], Nico Dissmeyer[2,3] & Emily Flashman[1]

Crop yield loss due to flooding is a threat to food security. Submergence-induced hypoxia in plants results in stabilization of group VII ETHYLENE RESPONSE FACTORs (ERF-VIIs), which aid survival under these adverse conditions. ERF-VII stability is controlled by the N-end rule pathway, which proposes that ERF-VII N-terminal cysteine oxidation in normoxia enables arginylation followed by proteasomal degradation. The PLANT CYSTEINE OXIDASEs (PCOs) have been identified as catalysts of this oxidation. ERF-VII stabilization in hypoxia presumably arises from reduced PCO activity. We directly demonstrate that PCO dioxygenase activity produces Cys-sulfinic acid at the N terminus of an ERF-VII peptide, which then undergoes efficient arginylation by an arginyl transferase (ATE1). This provides molecular evidence of N-terminal Cys-sulfinic acid formation and arginylation by N-end rule pathway components, and a substrate of ATE1 in plants. The PCOs and ATE1 may be viable intervention targets to stabilize N-end rule substrates, including ERF-VIIs, to enhance submergence tolerance in agriculture.

[1] Chemistry Research Laboratory, University of Oxford, 12 Mansfield Road, Oxford OX1 3TA, UK. [2] Independent Junior Research Group on Protein Recognition and Degradation, Leibniz Institute of Plant Biochemistry (IPB), Weinberg 3, D-06120 Halle (Saale), Germany. [3] ScienceCampus Halle Plant - based Bioeconomy, Betty-Heimann-Strasse 3, D-06120 Halle (Saale), Germany. [4] Institute of Biology I, RWTH Aachen University, Worringerweg 1, D-52074 Aachen, Germany. [5] Chemical Genomics Centre of the Max Planck Society, Otto-Hahn-Strasse 15, D-44227 Dortmund, Germany. [6] VU University Amsterdam, De Boelelaan 1083, 1081 HV Amsterdam, The Netherlands. [7] Department of Biochemistry, University of Oxford, South Parks Road, Oxford OX1 3QU, UK. Correspondence and requests for materials should be addressed to N.D. (email: nico.dissmeyer@ipb-halle.de) or to E.F. (email: emily.flashman@chem.ox.ac.uk).

All aerobic organisms require homeostatic mechanisms to ensure $O_2$ supply and demand are balanced. When supply is reduced (hypoxia), a hypoxic response is required to decrease demand and/or improve supply. In animals, this well-characterized response is mediated by the hypoxia-inducible transcription factor (HIF), which upregulates genes encoding for vascular endothelial growth factor, erythropoietin and glycolytic enzymes among many others[1–3]. Hypoxia in plants is typically a consequence of reduced $O_2$ diffusion under conditions of waterlogging or submergence, or inside of organs such as seeds, embryos or floral meristems in buds where the various external cell layers act as diffusion barriers. Although plants can survive temporary periods of hypoxia, flooding has a negative impact on plant growth and, if sustained, it can result in plant damage or death[4]. This has a major impact on crop yield; for example, flooding resulted in crop loss costing $3 billion in the United States in 2011 (ref. 5). As climate change results in increased severe weather events including flooding[4], strategies to address crop survival under hypoxic stress are needed to meet the needs of a growing worldwide population.

The response to hypoxia in rice, Arabidopsis, and barley is known to be mediated by the group VII ETHYLENE RESPONSE FACTORs (ERF-VIIs)[6–11]. It has been found that these transcription factors promote the expression of core hypoxia-responsive genes, including those encoding alcohol dehydrogenase and pyruvate decarboxylase that facilitate anaerobic metabolism[12,13]. Crucially, it was shown, initially in Arabidopsis, that the stability of the ERF-VIIs is regulated in an $O_2$-dependent manner via the Arg/Cys branch of the N-end rule pathway, which directs proteins for proteasomal degradation depending on the identity of their amino-terminal amino acid[14–16]. Thus, a connection between $O_2$ availability and the plant hypoxic response was identified[11,17,18]. The Arabidopsis ERF-VIIs are translated with the conserved N-terminal motif MCGGAI/VSDY/F (ref. 4) and co-translational N-terminal methionine excision, catalysed by Met amino peptidases[19,20], leaves an exposed N-terminal Cys, which is susceptible to oxidation[14–16]. N-terminally oxidized Cys residues (Cys-sulfinic acid or Cys-sulfonic acid, Supplementary Fig. 1) are then proposed to render the ERF-VII N termini substrates for arginyl transfer RNA transferase (ATE)-catalysed arginylation. The subsequent Nt-Arg-ERF-VIIs are candidates for ubiquitination by the E3 ligase PROTEOLYSIS6 (PRT6) (ref. 21), which promotes targeted degradation via the 26S proteasome. It has also been shown that degradation of ERF-VIIs by the N-end rule pathway can be influenced by NO, and that the ERF-VIIs play a role in plant NO-mediated stress responses[22,23].

The plant hypoxic response mimics the equivalent well-characterized regulatory system in animals, whereby adaptation to hypoxia is mediated by HIF. In normoxic conditions, HIF is hydroxylated at specific prolyl residues targeting it for binding to the von Hippel–Lindau tumour suppressor protein, the recognition component of the E3-ubiquitin ligase complex, which results in HIF ubiquitination and proteasomal degradation[1,3]. Thus, although not substrates for the N-end rule pathway of protein degradation, HIF levels are regulated by posttranslational modification resulting in ubiquitination, in a manner that is sensitive to hypoxia. HIF prolyl hydroxylation is catalysed by $O_2$-dependent enzymes, the HIF prolyl hydroxylases 1–3 (ref. 2), which are highly sensitive to $O_2$ availability[24,25]. These $O_2$-sensing enzymes are thus the direct link between $O_2$ availability and the hypoxic response[26,27].

Crucially, a family of five enzymes, the PLANT CYSTEINE OXIDASEs (PCO1–5), were identified in Arabidopsis[28] that were reported to catalyse the $O_2$-dependent reaction in the plant hypoxic response, specifically the oxidation of the conserved Cys residue at the N terminus of the Arabidopsis ERF-VIIs, RAP2.2, RAP2.12, RAP2.3, HRE1 and HRE2. It was found that overexpression of PCO1 and 2 in planta specifically led to depleted RAP2.12 protein levels and reduced submergence tolerance, whereas pco1 pco2 T-DNA insertion mutants accumulated RAP2.12 protein. Isolated recombinant PCO1 and PCO2 were shown to consume $O_2$ in the presence of pentameric peptides CGGAI corresponding to the N termini of various ERF-VIIs (Supplementary Table 1)[28]. The identification of these enzymes indicates that the hypoxic response in plants is enzymatically regulated[28], potentially in a similar manner to the regulation of the hypoxic response in animals by the HIF hydroxylases. The PCOs may therefore act as plant $O_2$ sensors.

Validation of the chemical steps in the Arg/Cys branch of the N-end rule pathway is still limited, both in animals and plants. We therefore sought to provide molecular evidence that the PCOs catalyse the oxidation step in ERF-VII proteasomal targeting and to determine whether this step is required for further molecular priming by arginylation. Using mass spectrometry (MS) and nuclear magnetic resonance (NMR) techniques, we confirm that PCO1 and also PCO4—representatives of the two different PCO 'subclasses' based on sequence identity and expression behaviour[28]—catalyse dioxygenation of the N-terminal Cys of Arabidopsis ERF-VII peptide sequences to Cys-sulfinic acid ($CysO_2$). This oxidation directly incorporates molecular $O_2$. To our knowledge, these are the first described enzymes that catalyse cysteinyl oxidation, as well as being the first described cysteine dioxygenases in plants. We then verify that the Cys-sulfinic acid product of the PCO-catalysed reactions is a direct substrate for the arginyl tRNA transferase ATE1, demonstrating that PCO activity is relevant and sufficient for the subsequent step of molecular recognition and modification according to the N-end rule pathway. This provides the first molecular evidence that Nt-Cys-sulfinic acid is a bona fide substrate for N-end rule-mediated arginylation. Overall, we thus define the PCOs as plant cysteinyl dioxygenases and ATE1 as an active arginyl transferase, establishing for the first time a direct link between molecular $O_2$, PCO catalysis and ATE1 recognition and modification of N-end rule substrates.

## Results

**PCOs catalyse $O_2$-dependent modification of RAP2$_{2-11}$.** N-terminally hexahistidine-tagged recombinant PCO1 and 4 were purified to ~90% purity, as judged by SDS–polyacrylamide gel electrophoresis (Supplementary Fig. 2a). Protein identity was confirmed by comparison of observed and predicted mass by liquid chromatography (LC)–MS (PCO1 predicted mass 36,510 Da, observed mass 36,513 Da; PCO4 predicted mass 30,680 Da, observed mass 30,681 Da, Supplementary Fig. 2b). Both PCO1 and PCO4 were found to be monomeric in solution and to co-purify with substoichiometric levels of Fe(II) (~0.3 atoms Fe(II) per monomer, Supplementary Fig. 2c–e), in line with the reported parameters of recombinant forms of their distant homologues, the cysteine dioxygenases (CDOs)[28–30]. The activity of the purified PCO1 and PCO4 was tested towards a synthetic 10-mer peptide corresponding to the methionine excised N termini of the ERF-VIIs RAP2.2, RAP2.12 and HRE2 ($H_2N$-CGGAIISDFI-COOH, hereafter termed RAP2$_{2-11}$ Supplementary Table 1). Assays comprising RAP2$_{2-11}$ at 100 μM in the presence or absence of PCO1 or PCO4 at 0.5 μM underwent aerobic or anaerobic coincubation for 30 min at 30 °C before analysis of the peptide by matrix-assisted laser desorption/ionization-MS (MALDI–MS, Fig. 1a,b). Only under aerobic conditions and in the presence of PCO1 or PCO4 did the spectra reveal the appearance of two species with mass increases of +32 Da and +48 Da, corresponding to two or three added

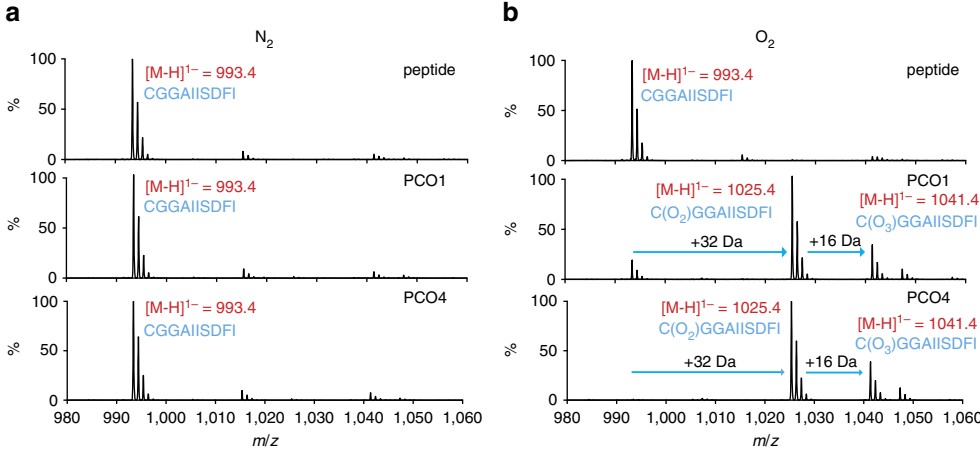

**Figure 1 | O$_2$-dependent Cys-modification of a RAP2$_{2-11}$ peptide substrate.** MALDI–MS spectra showing the RAP2$_{2-11}$ peptide species identified following incubation with PCO1 and PCO4 under anaerobic (**a**) or aerobic (**b**) conditions. Products with mass increases of +32 Da and +48 Da were only observed in the presence of PCO1 or PCO4 and O$_2$.

O atoms, suggesting an O$_2$-dependent reaction for PCOs 1 and 4 (Fig. 1b), as previously shown for PCOs 1 and 2 (it is noteworthy that supplementation of Fe(II) and/or addition of ascorbate was not required for the endpoint PCO1/4 activity assays conducted in this study)[28]. These mass shifts were deemed to be consistent with enzymatic formation of Cys-sulfinic (CysO$_2$, +32 Da) and Cys-sulfonic acid (CysO$_3$, +48 Da; Supplementary Fig. 1). Although homology between the PCOs and CDOs[28,30] leads to the predisposition that they will perform similar chemistry (that is, catalyse Cys sulfinic acid formation), both Cys-sulfinic and Cys-sulfonic acid are proposed to be Arg transferase substrates in the Arg/Cys branch of N-end rule-mediated protein degradation and therefore both were considered as potential products of the PCO-catalysed reaction[14–16].

**PCOs catalyse dioxygenation of RAP2$_{2-11}$.** To ascertain whether the PCOs function as dioxygenases and thus to confirm a direct connection between molecular O$_2$ and PCO activity, we sought to verify the source of the O atoms in the oxidized RAP2$_{2-11}$ by conducting assays in the presence of $^{18}$O$_2$ as the cosubstrate or H$_2^{18}$O as the solvent. To probe O$_2$ as the source of O atoms in the product, anaerobic solutions of RAP2$_{2-11}$ were prepared in sealed vials before addition of PCO4 using a gas-tight syringe. The vials were then purged with $^{16}$O$_2$ or $^{18}$O$_2$ and the reactions were allowed to proceed at 30 °C for a subsequent 20 min. Upon analysis by MALDI–MS, the mass of the products revealed that molecular O$_2$ was incorporated into the Cys-sulfinic acid product (Fig. 2a). The Cys-sulfinic acid product had a mass of +32 Da in the presence of $^{16}$O$_2$ and +36 Da in the presence of $^{18}$O$_2$, demonstrating addition of two $^{18}$O atoms and indicating that O$_2$ is the source of O atoms in this product. The Cys-sulfonic acid product had a mass of +52 Da in the presence of $^{18}$O$_2$, indicating a third $^{18}$O atom had not been incorporated into this product. To probe whether the source of the additional mass in the apparent Cys-sulfonic acid product was an O atom derived from water, an equivalent reaction was carried out under aerobic conditions in the presence of H$_2^{18}$O (H$_2^{18}$O:H$_2$O in a 3:1 ratio). No additional mass was observed in the peak corresponding to the Cys-sulfonic acid, raising the possibility that the +48 Da species observed by MALDI–MS is not enzymatically formed. Importantly, following incubation in the presence of H$_2^{18}$O, no additional mass was observed in the peak corresponding to Cys-sulfinic acid, confirming that this species is a product of a reaction where molecular O$_2$ is a substrate (Fig. 2b).

To further investigate whether the PCO-catalysed product species observed at +48 Da is enzymatically produced or an artefact of the MALDI–MS analysis method, we turned to LC–MS, to analyse the products of the PCO-catalysed reactions. Under these conditions, only peptidic product with a mass increase of +32 Da was observed after incubation with both PCO1 and PCO4, corresponding to the incorporation of two O atoms and the formation of Cys-sulfinic acid (Fig. 2c), consistent with the products observed using $^{18}$O$_2$ and H$_2^{18}$O (Fig. 2a,b). No product was observed with a mass corresponding to Cys-sulfonic acid, which suggested that the +48 Da product detected by MALDI–MS was indeed an artefact. When combined with the observation that significant quantities of Cys-sulfonic acid were not seen in the no-enzyme or anaerobic controls (Fig. 1), it was hypothesized that the Cys-sulfinic acid product of the PCO-catalysed reaction is non-enzymatically converted to Cys-sulfonic acid during MALDI–MS analysis, potentially as a result of laser exposure. Upon subjecting the products of PCO1 and 4 turnover to MALDI–MS analysis with increasing laser intensity, a direct correlation between laser intensity and the ratio of Cys-sulfonic acid:Cys-sulfinic acid product was observed (Supplementary Fig. 3a). Of note, significant levels of laser induced formation of +32 and +48 Da species upon analysis of unmodified peptide were not observed (Supplementary Fig. 3b). Together, these results confirm that the +48 Da species observed following incubation of the PCOs with RAP2$_{2-11}$ are a product of Cys-sulfinic acid exposure to the MALDI–MS laser and not a product of the PCO-catalysed reaction. Overall, these data demonstrate that the PCOs are dioxygenase enzymes, similar to the mammalian and bacterial CDOs to which they show sequence homology[28,30].

**PCOs catalyse RAP2$_{2-11}$ N-terminal Cys-sulfinic acid formation.** Recombinant PCO1 and PCO2 were reported to consume O$_2$ in the presence of pentameric CGGAI peptides corresponding to the methionine-excised N terminus of the *Arabidopsis* ERF-VIIs[28]. To definitively verify that the N-terminal cysteinyl residue of RAP2$_{2-11}$ is indeed the target for the PCO-catalysed +32 Da modifications, we conducted LC–MS/MS analyses on the reaction products. Fragmentation of RAP2$_{2-11}$ that had been incubated in the presence and absence of PCO1 and PCO4 revealed *b*- and *y*-ion series consistent with oxidation of the N-terminal Cys residue (Fig. 3a), confirming that PCOs 1 and 4 act as cysteinyl dioxygenases.

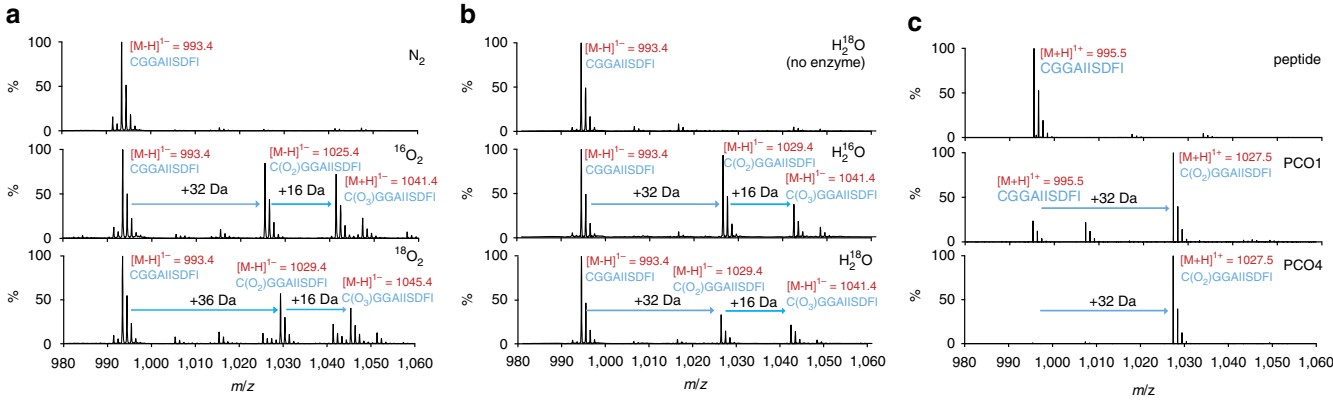

**Figure 2 | PCOs catalyse incorporation of molecular O$_2$ into RAP2$_{2-11}$.** (**a**) MALDI–MS spectra showing that PCO4-catalysed reactions carried out in the presence of $^{18}O_2$ result in a $+4$ Da increase in the mass of the putative Cys-sulfinic acid product; however, a $+6$ Da increase in the size of the putative Cys-sulfonic acid product is not observed; (**b**) MALDI–MS spectra showing that PCO4-catalysed reactions carried out in the presence of $H_2^{18}O$ show no additional incorporation of mass compared with products of reactions in the presence of $H_2^{16}O$; (**c**) LC–MS spectra confirm that the $+48$ Da reaction product is an artefact of MALDI–MS analysis (Supplementary Fig. 3) and incubation of PCO1 and PCO4 with RAP2$_{2-11}$ results in formation of a single product with a mass increase of $+32$ Da, consistent with Cys-sulfinic acid formation.

As a final confirmation of the nature of the reaction catalysed by PCO1 and PCO4, their activity was monitored using $^1$H-NMR. Reactions were initiated by adding 5 μM enzyme to 500 μM RAP2$_{2-11}$ (in the presence of 10% $D_2O$) and products of the reaction were analysed using a 600 MHz NMR spectrometer. In the presence of both PCO1 and PCO4, modification to the cysteinyl residues was observed, as exemplified by the disappearance of the $^1$H-resonance corresponding to the β-cysteinyl protons (at $\delta_H$ 2.88 p.p.m.) and the emergence of a new $^1$H-resonance at $\delta_H$ 2.67 p.p.m. (Fig. 3b). The chemical shift of the new resonance is similar to that observed for L-Cys conversion to L-Cys-sulfinic acid by mouse CDO (ref. 31) and also to the chemical shift of an L-Cys-sulfinic acid standard measured under equivalent conditions to the PCO assays (Supplementary Fig. 4). Therefore, the resonance shift observed upon PCO1/4 reaction was assigned to the β-protons of L-Cys-sulfinic acid. Overall, these results provide confirmation at the molecular level that *Arabidopsis* PCOs 1 and 4 act as plant cysteinyl dioxygenases, catalysing incorporation of $O_2$ into N-terminal Cys residues on a RAP2 peptide to form Cys-sulfinic acid.

**ATE1 arginylates acidic N termini including Cys-sulfinic acid.** We next sought to confirm that the PCO-catalysed Cys-oxidation to Cys-sulfinic acid renders a RAP2 peptide capable of and sufficient for onward modification by ATE1. Cys-sulfinic acid has been proposed as a substrate for ATE1 on the basis of its structural homology with known ATE1 substrates Asp and Glu, but evidence has only been reported to date for arginylation of Cys-sulfonic acid[32,33]. We further sought to validate the role of a plant ATE1: to date ATE1 has been suggested to be responsible for transfer of $^3$H-arginine to bovine α-lactalbumin in highly purified plant extracts *in vitro*[34] and RAP2.12 stabilization in *ate1 ate2* double-null mutant plant lines implicates ATE1 as an ERF-VII-targeting arginyl transferase *in vivo*[17,18]. To this end, we produced recombinant hexahistidine-tagged *Arabidopsis* ATE1 (Supplementary Fig. 5) for use in an arginylation assay, which detects incorporation of radiolabelled $^{14}$C-Arg into biotinylated peptides. Carboxy-terminally biotinylated RAP2$_{2-13}$ peptides (H$_2$N-*X*GGAIISDFIPP(PEG)K(biotin)-NH$_2$) where the N-terminal residue, *X*, constitutes Gly, Asp, Cys or Cys-sulfonic acid were subjected to the arginylation assay in the presence or absence of PCO1/4 (Fig. 4a). Peptide with an N-terminal Gly did

not accept Arg, whereas an N-terminal Asp did accept Arg, independent of the presence of PCO1 or 4. A peptide comprising an N-terminal Cys-sulfonic acid was also shown to be a substrate for ATE1, again independent of the presence of PCO1 or 4, which is in line with proposed steps of the Arg/Cys N-end rule pathway and has also recently been reported using a similar assay with mouse ATE1 (refs 14–16,35). Crucially, in the absence of PCO1/4, RAP2$_{2-13}$ with an N-terminal Cys was not an acceptor of arginine transfer by ATE1, yet when either PCO1 or PCO4 was incorporated in the reaction, significant ATE1 transferase activity was observed (Fig. 4a).

To confirm that the increased detection of radiolabelled arginine corresponded to arginyl incorporation at the N termini of the peptides, the experiment was repeated using non-radiolabelled arginine in the presence and absence of PCO4 and ATE1, and peptide products subjected to LC–MS analysis (Fig. 4c). As with RAP2$_{2-11}$ (Fig. 2c), the Cys-initiated RAP2$_{2-13}$ peptide displayed a $+32$ Da increase in mass upon incubation with PCO4 only (Fig. 4c, red spectrum). Importantly, following incubation of Cys-initiated RAP2$_{2-13}$ with both PCO4 and ATE1, a mass increase equivalent to oxidation coupled to arginylation ($+188$ Da) was observed (Fig. 4c, blue spectrum). Subsequent tandem MS analysis of these product ions revealed fragmentation species consistent with the assumption that oxidation and sequential arginylation occur at the N terminus of PCO4- and ATE1-treated peptides (Fig. 4d, blue spectrum), strongly suggesting that the PCO-oxidized N termini of ERF-VIIs are rendered N-degrons via additional arginylation (Fig. 4b).

A $+12$ Da mass increase was observed in control assays lacking PCO4 (Fig. 4c,d; purple spectra). This appeared to be related to prolonged incubation in the presence of HEPES and dithiothreitol (DTT) as used in the arginylation assay buffer: The $+12$ Da modification was not observed if the peptide was dissolved in $H_2O$ (Fig. 4c, black spectrum) or if incubated with HEPES and DTT for just 1 h, but was observed when the peptide was incubated with HEPES and DTT overnight (Supplementary Fig. 6). It is proposed that under these conditions, trace levels of contaminating formaldehyde react with free Nt-Cys residues to form thiazolidine N termini[36].

These results are in line with proposed arginylation requirements for the Arg/Cys branch of the N-end rule pathway[14–16] including the known Cys-initiated arginylation targets from mammals[32,33,35,37]. Importantly, these results demonstrate for the

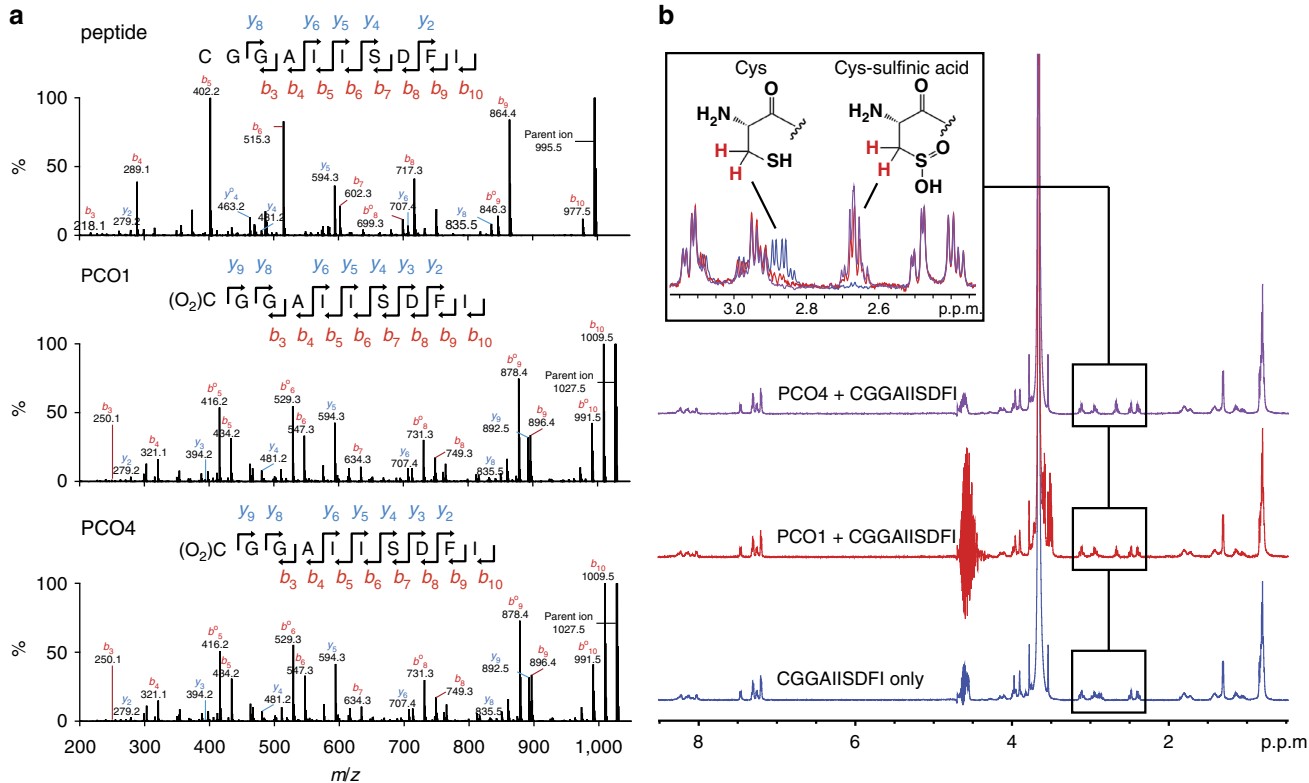

**Figure 3 | PCO1 and PCO4 oxidize the N-terminal Cys of RAP2$_{2-11}$ to Cys-sulfinic acid. (a)** Peptidic products of PCO-catalysed reactions were subjected to LC–MS/MS analysis. In the presence of enzyme, fragment assignment was consistent with expected b- and y-series ion masses for RAP2$_{2-11}$ with N-terminal Cys-sulfinic acid. **(b)** $^1$H-NMR was used to monitor changes to RAP2$_{2-11}$ (500 μM) upon incubation with enzyme (5 μM). In the presence of PCO1 (red) and PCO4 (purple), the $^1$H-resonance at $\delta_H$ 2.88 p.p.m. (assigned to the β-cysteinyl protons of RAP2$_{2-11}$, blue) was observed to decrease in intensity, with concomitant emergence of a resonance at $\delta_H$ 2.67 p.p.m. This new resonance was assigned to the β-protons of Cys-sulfinic based on chemical shift analysis (see Supplementary Fig. 4).

first time Arg transfer mediated by a plant ATE dependent on the N-terminal residue of its substrate, and also that both Cys-sulfinic acid (the product of PCO-catalysis) and Cys-sulfonic acid can act as substrates for ATE1. In particular, the arginylation observed with PCO-catalysed Cys-sulfinic acid supports the assumption that N-terminal residues sterically and electrostatically resembling Asp or Glu can serve as Arg acceptors in reactions catalysed by ATEs[33], and also confirms the importance of the PCOs as a connection between the stability of their ERF-VII substrates and O$_2$ availability (Fig. 4b).

## Discussion

The PCOs were identified in *Arabidopsis* as a set of five enzymes suggested to catalyse oxidation of N-terminal cysteine residues in ERF-VII transcription factors and oxygen consumption was demonstrated for reactions with short peptides corresponding to their N termini[28]. This putative oxidation was associated with destabilization of the ERF-VIIs, presumably by rendering them substrates of the Arg/Cys branch of the N-end rule pathway[14,16]. Under conditions of sufficient O$_2$ availability ERF-VII protein levels are decreased, whereas under hypoxic conditions, such as those encountered upon plant submergence or in the context of organ development, ERF-VII levels remain high[17,18]. Importantly, the ERF-VII transcription factors are known to upregulate genes which allow plants to cope with or respond to submergence[13]. The PCOs are proposed to act as potential O$_2$ sensors involved in regulating the plant hypoxic response[28].

We sought to biochemically confirm the role of the PCOs in the plant hypoxic response, and present here MS and NMR data

that clearly demonstrate that two enzymes from different 'subclasses' of this family, PCOs 1 and 4, are dioxygenases that catalyse direct incorporation of O$_2$ into RAP2$_{2-11}$ peptides to form Cys-sulfinic acid. Their direct use of O$_2$ supports the proposal that these enzymes may act as plant O$_2$ sensors[28]. A relationship has been demonstrated between O$_2$ concentration and PCO activity[28], but it will be of interest to perform detailed kinetic characterization of these enzymes to ascertain their level of sensitivity to O$_2$ availability, in particular to determine whether their O$_2$-sensitivity is similar to that of the HIF hydroxylases in animals[24,25]. Although there is functional homology between the PCOs and the HIF hydroxylases, they are apparently mechanistically divergent: the PCOs show sequence homology to the Fe(II)-dependent CDO family of enzymes, which do not require an external electron donor for O$_2$ activation[28,30], whereas the HIF hydroxylases are Fe(II)/2OG-dependent oxygenases. They also co-purified with Fe(II) as reported for both the CDOs[29] and prolyl hydroxylase 2 (ref. 38). Of note, the PCOs are the first identified CDOs in plants. Further, in contrast to the reactions of mammalian and bacterial CDOs, which oxidize free L-Cys, the PCOs are also, to our knowledge, the first identified cysteinyl (as opposed to free L-Cys) dioxygenases.

According to the Arg/Cys branch of the N-end rule pathway, N-terminal Cys oxidation is proposed to enable successive arginylation by ATE1 to render proteins as N-degrons. Although both Cys-sulfinic and Cys-sulfonic acid are repeatedly reported as potential arginylation substrates[14–16], detailed evidence has only been presented to date for arginylation of Cys-sulfonic acid[32,33] and this only in a mammalian system. We therefore sought to

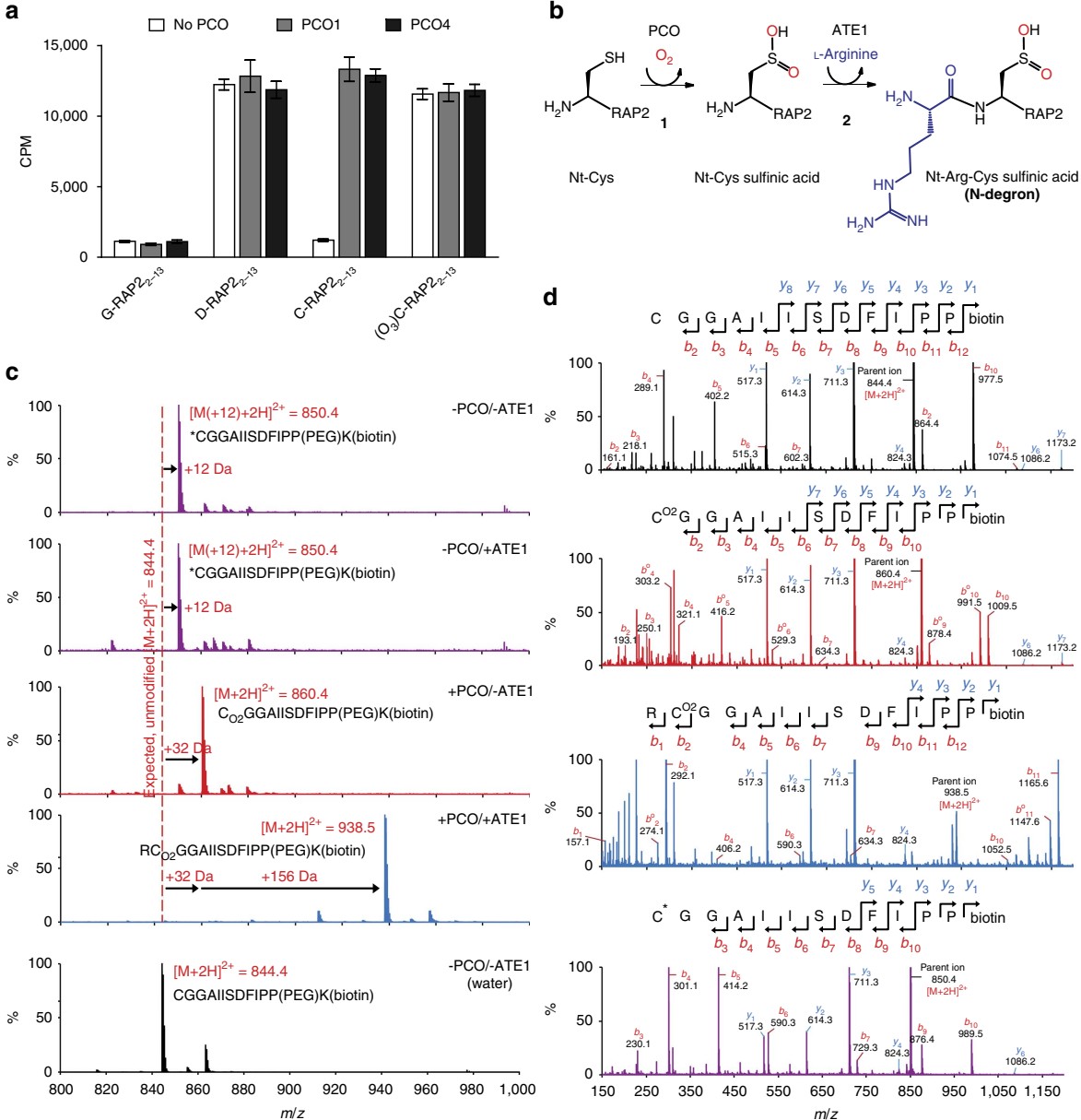

**Figure 4 | PCO-catalysed Cys-sulfinic acid formation renders RAP2$_{2-13}$ a substrate for ATE1-catalysed arginylation. (a)** $^{14}$C-Arg incorporation by ATE1 into the 12-mer N-terminal ERF-VII peptide (H$_2$N-$X$GGAIISDFIPP(PEG)K(biotin)-NH$_2$, $X$ = Gly, Asp, Cys or Cys-sulfonic acid (C(O$_3$))), was assayed by liquid scintillation counting of immobilized biotinylated peptides after the arginylation reaction and removal of unreacted $^{14}$C-Arg ($n = 3$). In the case of the Cys-starting peptide (RAP2$_{2-13}$), ATE1 activity was strongly dependent on the presence of PCO1 or PCO4. $n = 3$, error bars in this panel represent s.e.m. **(b)** Scheme showing PCO- and ATE1-catalysed reactions on Nt-Cys of ERF-VIIs, as validated in this study. **(c)** LC–MS spectra of products of equivalent assays with Cys-initiated RAP2$_{2-13}$ using non-radiolabelled Arg, revealing a sequential mass increase of $+32$ (corresponding to oxidation) and $+156$ Da (corresponding to arginylation) only in the presence of PCO and ATE1 (blue spectrum). The red spectrum shows a $+32$ Da mass increase for Cys-RAP2$_{2-13}$ incubated $+$PCO/$-$ATE, demonstrating Cys-sulfinic acid formation as expected. Purple spectra show $+12$ Da species formed upon incubation of Cys-RAP2$_{2-13}$ in the absence of PCO $+/-$ ATE (for explanation of this mass increase see text and Supplementary Fig. 6); the black spectrum shows Cys-RAP2$_{2-13}$ dissolved in H$_2$O. **(d)** $b$- and $y$-ion series spectra generated by MS/MS analysis of Cys-RAP2$_{2-13}$ only (no incubation; black), Cys-RAP2$_{2-13}$ incubated $+$PCO/$-$ATE (red), Cys-RAP2$_{2-13}$ incubated $+$PCO/$+$ATE1 (blue) and Cys-RAP2$_{2-13}$ incubated $-$PCO/$-$ATE (purple), confirming arginylation only at the N terminus of PCO-modified RAP2$_{2-13}$.

demonstrate that PCO-catalysed ERF-VII N-terminal Cys oxidation to Cys-sulfinic acid promotes arginylation by ATE1. The arginylation assay and MS results we present demonstrate that the PCO-catalysed dioxygenation reaction is sufficient to trigger N-terminal arginylation of ERF-VIIs by ATE1, thus probably rendering N-termini of ERF-VIIs (at least those comprising the tested N-terminal sequence) as N-degrons, that is, allowing recognition by PRT6 and other potential E3 ubiquitin

ligases, polyubiquitination and possibly transfer to the 26S proteasome for proteolysis[14–16]. Collectively, therefore, we present the comprehensive molecular evidence confirming the Cys oxidation and subsequent arginylation steps of the Arg/Cys branch of the N-end rule pathway[32,33,37]. We also confirm that ATE1 is able to selectively arginylate, as predicted[33], acidic N-terminal residues of plant substrates, including Cys-sulfonic acid.

Arginylation has been known as a post-translational modification since 1963 (ref. 39), to possess a general aminoacyl transferase function in plants (rice and wheat) since 1973 (ref. 40) and to have a speculative involvement in the N-end rule pathway since 1988 (refs 41,42). ATE1 is reported as being capable of arginylating proteins at both acidic N termini and midchain acidic side chains via canonical and non-canonical peptide bonds, respectively[43]. Reports of midchain arginylation highlighted a potentially broad involvement of ATEs in posttranslational protein modifications for various functions[35,43–45] but was only very recently brought into question by [14]C-Arg incorporation assays using arrays of immobilized synthetic peptides[35,43,44]. However, to date only one physiological and two in vitro substrates for the Arg/Cys branch of the N-end rule pathway have been characterized, namely mammalian regulator of G-protein signalling (RGS) 4, and RGS5 and 10, respectively[46], where Nt-Cys oxidation was described (to Cys-sulfonic acid) as was Nt-Cys arginylation[33,37]. The first non Cys-branch N-end rule arginylation target was shown to require posttranslational proteolytic cleavage of a (pre-)-proprotein. The C-terminal fragment of proteolytically cleaved mouse BRCA1 is Asp-initiated[47] and gets degraded in an N-end rule-dependent manner. Then, the molecular chaperone BiP (GRP78 and HSPA5, heat shock 70 kDa protein 5) and the oxidoreductase protein disulphide isomerase, present Glu or Asp after cleavage of their signal peptide, respectively, and were suggested but not shown as putative N-end rule substrates[48]. Only very recently, BiP and protein disulphide isomerase were identified in mammalian cell culture together with the Glu-initiated calreticulin as arginylation targets with a function in autophagy rather than the N-end rule degradation[49].

Similarly, data regarding the molecular requirements of plant ATEs are limited. Already in 1973, a general aminoacyl transfer activity was found in rice and wheat cell extracts, however, the nature of enzyme, acceptor position and mechanism remained unclear. It was suggested that the N-terminus could serve as Arg acceptor[40].

The first description of a mutant of the single translatable ATE1 gene in the Arabidopsis accession Wassilewskija (Ws-0) highlighted a role of ATEs in plant development. Ws-0 lacks the second bona fide ATE, that is, ATE2, due to a single-nucleotide polymorphism in ATE2 causing a premature stop[50]. Developmental functions of the single homologue ATE1 in the moss Physcomitrella patens were recently described[51]. Interaction partners of the enzyme were found as well as four arginylated peptides immunologically detected by using antibodies directed against peptides mimicking N-terminal Arg-Asp or Arg-Glu[52]. In one case, that is the acylamino-acid-releasing enzyme PpAARE, which presents for unknown reasons a neo-N-terminal Asp residue, which was formerly Asp2 and therefore initiated by Met, an N-terminal arginylation was found with high confidence. Previously, Arg transferase function of Arabidopsis ATE1/2 has been shown using an assay detecting conjugation of [3]H-Arg to bovine α-lactalbumin (bearing an N-terminal Glu) in the presence of plant extracts from wild-type Arabidopsis, and ate1 and ate2 single mutants but not from ate1 ate2 double mutant seedlings[34]. Therefore, the results we present here demonstrate for the first time Arg transferase activity of a plant ATE towards known plant N-end rule substrates.

Interestingly, in combination with $O_2$, nitric oxide was identified as an RGS-oxidizing agent, suggesting a potential role of S-nitrosylation in the Arg/Cys branch of the N-end rule pathway, albeit non-enzymatically controlled[32]. It has also been reported in planta that both NO and $O_2$ are required for ERF-VII degradation, potentially at the Cys oxidation step[22,23]. Although in N-end rule-mediated RGS4/5 degradation it has

been proposed that Cys nitrosylation precedes Cys oxidation (also currently considered a non-enzymatic process), we find that under the conditions used, the PCO1/4-catalysed reaction does not require either prior Cys nitrosylation or exogenous NO to proceed efficiently. We cannot rule out that NO plays a role in formation of a Cys-sulfonic acid product, which is also a substrate for ATE1 as shown in our Arg transfer experiments. Alternatively, NO may have a role in ERF-VII degradation in vivo via non-enzymatic oxidation or via a secondary mechanism. The manner in which NO contributes to Arg/Cys branch of the N-end rule pathway therefore remains to be elucidated.

ERF-VII stabilization has been shown to result in improved submergence tolerance, elegantly demonstrated in barley by mutation of the candidate E3-ubiquitin ligase PRT6 (ref. 11), but also in rice containing the Sub1A gene; SUB1A is an apparently stable ERF-VII that confers particular flood tolerance in certain rare varieties of rice[9,17]. Overexpression of Sub1A in more commonly grown rice varieties has resulted in a 45% increase in yield relative to sub1a mutant lines after exposure to flooding[53]. If ERF-VII stabilization is indeed a proficient mechanism for enhancing flood tolerance, then manipulation of PCO or ATE activity may be an efficient and effective point of intervention. This work presents molecular validation of their function, providing the basis for future targeted chemical/genetic inhibition of their activity. It also highlights genetic strategies for breeding via introgression of variants of N-end rule pathway components or introduction of alleles of enzymatic components of the N-end rule pathway from non-crop species into crops. Any of these strategies has the potential to result in stabilized ERF-VII levels and increase stress resistance, and may therefore help to address food security challenges.

## Methods

**Peptide synthesis.** All reagents used were purchased from Sigma-Aldrich unless otherwise stated. The 10-mer RAP2$_{2-11}$ peptide (H$_2$N-CGGAIISDFI-COOH) was purchased from GL Biochem (Shanghai) Ltd, China (Supplementary Table 1). The sequence of the 12-mer RAP2$_{2-13}$ peptides used in the coupled oxidation-arginylation assay is derived from RAP2.2, RAP2.12 and HRE2 (H$_2$N-X-GGAIISDFIPP(PEG)K(biotin)-NH$_2$), and synthesized by Fmoc-based solid-phase peptide synthesis on NovaSyn TGR resin (Merck KGaA, Supplementary Table 2). Fmoc protected amino acids (Iris Biotech GmbH) were coupled using 4 equivalents (eq) of the amino acid according to the initial loading of the resin. 4 eq amino acid was mixed with 4 eq O-(6-chlorobenzotriazol-1-yl)-N,N,N′,N′-tetramethyluronium hexafluorophosphate and 8 eq N,N-diisopropy-lethylamine (Santa Cruz Biotechnology, sc-293894), and added to the resin for 1 h. In a second coupling, the resin was treated with 4 eq of the Fmoc-protected amino acid mixed with 4 eq benzotriazole-1-yl-oxy-tris-pyrrolidino-phosphonium hexafluoro-phosphate and 8 eq 4-methylmorpholine for 1 h. After double coupling a capping step to block free amines was performed using acetanhydride and N,N-diisopropylethylamine in N-methyl-2-pyrrolidinone (1:1:10) for 5 min. The C-terminal Fmoc-Lys(biotin)-OH, the 8-(9-fluorenylmethyloxycarbonyl-amino)-3.6-dioxaoctanoic acid (PEG) linker and the different Fmoc protected N-terminal amino acids were coupled manually. The remaining peptide sequence was assembled using an automated synthesizer (Syro II, MultiSynTech GmbH). Fmoc deprotection was performed using 20% piperidine in dimethylformamide (DMF) for 5 min, twice. After each step the resin was washed five times with DMF, methylene chloride (DCM) and DMF, respectively. Final cleavage was performed with 94% trifluoroacetic acid (TFA), 2.5% 1,2-ethanedithiole and 1% triisopro-pylsilane in aqueous solution for 2 h, twice. The cleavage solutions were combined and peptides were precipitated using diethyl ether (Et$_2$O) at − 20 °C for 30 min. Peptides were solved in water/acetonitrile (ACN) 7:3 and purified by reversed-phase HPLC (Nucleodur C18 culumn; 10 × 125 mm, 110 Å, 5 μm particle size; Macherey-Nagel) using a flow rate of 6 ml min$^{-1}$ (A: ACN with 0.1% TFA, B: water with 0.1% TFA). Obtained pure fractions were pooled and lyophilized. Peptide characterization was performed by analytical HPLC (1260 Infinity, Agilent Technology; flow rate of 1 ml min$^{-1}$, A: ACN with 1% TFA, B: water with 1% TFA) coupled with a mass spectrometer (6120 Quadrupole LC–MS, Agilent Technology) using electrospray ionization (Agilent Eclipse XDB-C18 culumn, 4.6 × 150 mm, 5 μm particle size). Analytical HPLC chromatograms were recorded at 210 nm (Supplementary Fig. 7). Quantification was performed by HPLC-based comparison (chromatogram at 210 nm) with a reference peptide (Supplementary Table 2).

**Protein expression and purification.** *Arabidopsis* PCO1 and PCO4 sequences in pDEST17 bacterial expression vectors (Invitrogen) were kindly provided by F. Licausi and J. van Dongen[28]. Plasmids were transformed into BL21(DE3) *Escherichia coli* cells and expression of recombinant protein carrying an N-terminal hexahistidine tag was induced with 0.5 mM isopropyl-β-D-thiogalactoside and subsequent growth at 18 °C for 18 h. Harvested cells were lysed by sonication and proteins purified using $Ni^{++}$ affinity chromatography, before buffer exchange into 250 mM NaCl/50 mM Tris-HCl (pH 7.5). Analysis by SDS–PAGE and denaturing LC–MS showed proteins with >90% purity and with the predicted molecular weights.

The coding sequence of *Arabidopsis* ATE1 was cloned according to gene annotations at TAIR (www.arabidopsis.org) from complementary DNA. The sequence was flanked by an N-terminal tobacco etch virus recognition sequence for facilitated downstream purification ('tev': ENLYFQ-X) using the primers ate1_tev_ss (5′-GCTTAGAGAATCTTTATTTTCAGGGGATGTCTTTGAAAAA CGATGCGAGT-3′) and ate1_as (5′-GGGGACCACTTTGTACAAGAAAGCTGG GTATCAGTTGATTTCATACACCATTCTCTC-3′). A second PCR using the primers adapter (5′-GGGGACAAGTTTGTACAAAAAAGCAGGCTTAGAGAAT CTTTATTTTCAGGGG-3′) and ate1_as was performed to amplify the construct to use it in a BP reaction for cloning into pDONR201 (Invitrogen) followed by an LR reaction into the vector pDEST17 (Invitrogen). The N-terminal hexahistidine fusion was expressed in BL21-CodonPlus (DE3)-RIL *E. coli* cells. The expression culture was induced with 1 mM isopropyl-β-D-thiogalactoside at optical density 0.6 and grown for 16 h at 18 °C. After resuspension in LEW buffer (50 mM $NaH_2PO_4$ pH 8, 300 mM NaCl and 1 mM DTT), the cells were lysed by incubation with 1.2 mg ml$^{-1}$ lysozyme for 30 min and underwent subsequent sonication in the presence of 1 mM phenylmethylsulfonyl fluoride. Recombinant protein was purified by $Ni^{++}$ affinity chromatography and subjected to Amicon Ultra-15 (30 K) (Merck Millipore) filtration for buffer exchange to imidazole-free LEW containing 20% glycerol.

**PCO activity assays and MALDI analysis.** PCO activity assays were conducted under the following conditions, unless otherwise stated: PCO1 or 4 (1 μM) was mixed with 100 or 200 μM $RAP2_{2-11}$ peptide in 250 mM NaCl, 1 mM DTT, 50 mM Tris-HCl pH 7.5 and incubated at 30 °C for 30–60 min. Addition of exogenous Fe(II) and/or ascorbate were not required for activity. Assays were stopped by quenching 1 μl sample with 1 μl α-cyano-4-hydroxycinnamic acid matrix on a MALDI plate before product mass analysis using a Sciex 4800 TOF/TOF mass spectrometer (Applied Biosystems) operated in negative ion reflectron mode. The instrument parameters and data acquisition were controlled by 4000 Series Explorer software and data processing was completed using Data Explorer (Applied Biosystems).

To test the activity of PCO4 in the presence of $^{18}O_2$, 100 μl of an anaerobic solution of 100 μM $RAP2_{2-11}$ in 250 mM NaCl/50 mM Tris-HCl pH 7.5 was prepared in a septum-sealed glass vial by purging with 100% $N_2$ for 10 min at 100 ml min$^{-1}$ using a mass flow controller (Brooks Instruments), as used for previous preparation of anaerobic samples to determine enzyme dependence on $O_2$ (ref. 24). PCO4 was then added using a gas-tight Hamilton syringe, followed by purging with a balloon ($\sim$0.7 l) of $^{16}O_2$ or $^{18}O_2$ over the course of 10 min at room temperature. Reaction vials were then transferred to 30 °C for a further 20 min before products were analysed by MALDI–MS as described above.

PCO4 activity was additionally tested in the presence of $H_2^{18}O$ by conducting an assay in 75% $H_2^{18}O$, 25% $H_2O$ (with all enzyme/substrate/buffer components comprising a portion of the $H_2O$ fraction). Assays were conducted for 10 min at room temperature followed by 20 min at 30 °C for comparison with assays conducted with $^{18}O_2$. Products were analysed by MALDI–MS, as described above.

**UPLC–MS and MS/MS analysis.** Ultra-high performance chromatography (UPLC)–MS measurements were obtained using an Acquity UPLC system coupled to a Xevo G2-S Q-ToF mass spectrometer (Waters) operated in positive electrospray mode. Instrument parameters, data acquisition and data processing were controlled by Masslynx 4.1. Source conditions were adjusted to maximize sensitivity and minimize fragmentation while Lockspray was employed during analysis to maintain mass accuracy. Two microlitres of each sample was injected on to a Chromolith Performance RP-18e 100-2 mm column (Merck) heated to 40 °C and eluted using a gradient of 95% deionized water supplemented with 0.1% (v/v) formic acid (analytical grade) to 95% acetonitrile (HPLC grade) and a flow rate of 0.3 ml min$^{-1}$. Fragmentation spectra of substrate and product peptide ions (MS/MS) were obtained using a targeted approach with a typical collision-induced dissociation energy ramp of 30 to 40 eV. Analysis was carried out with the same source settings, flow rate and column elution conditions as above.

**$^1$H-NMR assay.** Reaction components (5 μM PCO1 or PCO4 and 500 μM $RAP2_{2-11}$) were prepared to 75 μl in 156 mM NaCl, 31 mM Tris-HCl (pH 7.5) and 10% $D_2O$ (enzyme added last), in a 1.5 ml microcentrifuge tube before being transferred to a 2 mm diameter NMR tube. $^1$H-NMR spectra at 310 K were recorded using a Bruker AVIII 600 (with inverse cryoprobe optimized for $^1$H observation and running topspin 2 software; Bruker) and reported in p.p.m.

relative to $D_2O$ ($\delta_H$ 4.72). The deuterium signal was also used as internal lock signal and the solvent signal was suppressed by presaturating its resonance.

**Arginylation assay.** The conditions for arginylation of the 12-mer peptide substrates were modified from ref. 43. In detail, ATE1 was incubated at 10 μM in the reaction mixture containing 50 mM HEPES pH 7.5, 25 mM KCl, 15 mM $MgCl_2$, 1 mM DTT, 2.5 mM ATP; 0.6 mg ml$^{-1}$ *E. coli* tRNA (R1753, Sigma), 0.04 mg ml$^{-1}$ *E. coli* aminoacyl-tRNA synthetase (A3646, Sigma), 80 μM (4 nCi μl$^{-1}$) $^{14}C$-arginine (MC1243, Hartmann Analytic), 50 μM C-terminally biotinylated 12-mer peptide substrate and, where indicated, 1 μM purified recombinant PCO1 or PCO4 in a total reaction volume of 50 μl. The reaction was conducted at 30 °C for 16–40 h. After incubation, each 50 μl of avidin agarose bead slurry (20219, Pierce) equilibrated in PBSN (PBS-Nonidet; 100 mM $NaH_2PO_4$, 150 mM NaCl; 0.1% Nonidet-P40) was added to the samples and mixed with an additional 350 μl of PBSN. After 2 h of rotation at room temperature, the beads were washed four times in PBSN, resuspended in 4 ml of FilterSafe scintillation solution (Zinsser Analytic) and scintillation counting was performed using a Beckmann Coulter LS 6500 Multi-Purpose scintillation counter.

**Data availability.** The authors declare that all data supporting the findings of this study are available within the manuscript and its Supplementary Information files or are available from the corresponding authors upon request.

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

## Acknowledgements

Petra Majovsky, Domenika Thieme and Wolfgang Hoehenwarter from the Proteomics Unit of the Leibniz Institute of Plant Biochemistry (IPB), Halle, are acknowledged for MS of recombinant ATE1. David Staunton from the Biophysical Facility, Department of Biochemistry, University of Oxford, is acknowledged for multi-angle light scatter analysis of PCO1/4. Geoff Grime from the University of Surrey Ion Beam Centre is acknowledged for assistance with the microPIXE data collection. This work was supported by a Biotechnology and Biological Sciences Research Council (U.K.) New Investigator grant (BB/M024458/1) to E.F., a grant for setting up the junior research group of the ScienceCampus Halle–Plant-based Bioeconomy to N.D., by a PhD fellowship of the Landesgraduiertenförderung Sachsen-Anhalt awarded to C.N., by an Engineering and Physical Sciences Research Council (U.K.) studentship (EP/G03706X/1) to J.C.B.-B., a Royal Society Dorothy Hodgkin Fellowship to E.F., a William R. Miller Junior Research Fellowship (St Edmund Hall, Oxford) to R.J.H. and grant DI 1794/3-1 by the German Research Foundation (Deutsche Forschungsgemeinschaft, DFG) to N.D. Financial support came from the Leibniz Association, the state of Saxony Anhalt, the Deutsche Forschungsgemeinschaft (DFG) Graduate Training Center GRK1026 'Conformational Transitions in Macromolecular Interactions' at Halle, and the Leibniz Institute of Plant Biochemistry (IPB) at Halle, Germany. We thank Professor J. van Dongen (RWTH Aachen University, Germany) and Professor F. Licausi (Scuolo Superiore Sant'Anna, Pisa, Italy) for sharing pDEST-PCO plasmids and helpful discussions. The manuscript was deposited as pre-print before publication (https://doi.org/10.1101/069336) at bioRxiv—the preprint server for biology, operated by Cold Spring Harbor Laboratory (biorxiv.org). Publication of this article was funded by the Open Access fund of the Leibniz Institute of Plant Biochemistry (IPB). This work was supported by the network of the European Cooperation in Science and Technology (COST) Action BM1307—'European network to integrate research on intracellular proteolysis pathways in health and disease (PROTEOSTASIS)'.

## Author contributions

M.D.W. performed the PCO1/4 activity assays and MALDI/LC–/MS/MS analyses. M.K. performed and established arginylation reactions on peptides coupled to biotin pulldown and scintillation measurements and purified ATE1 protein. R.J.H. performed the NMR assays with E.F., D.A.W. prepared the pDEST17-PCO1 and four plasmids. C.M. synthesized the biotinylated peptides, T.N.G. supervised and designed the synthesis and C.N. cloned and established purification and activity assays for ATE1. R.O. conducted LC–MS to analyse +12 Da mass shifts. J.W. performed LC–MS analysis. J.Y. and J.C.B.-B. prepared samples for micro-PIXE analysis, and J.C.B.-B. and E.F.G. collected and analysed micro-PIXE data. E.F. performed the PCO1 and PCO4 protein purification and selected activity assays. E.F., M.D.W., M.K. and N.D. designed the study. E.F. and N.D. wrote the manuscript. M.D.W., M.K., N.D. and E.F. designed the figures. All authors read and approved the final version of this manuscript.

## Additional information

**Competing financial interests:** The authors declare no competing financial interests.

**Publisher's note**: 

