## [Peer Review File · Nature Communications]

Reviewers' comments:

Reviewer #1 (Remarks to the Author):

Overall this is an interesting paper presenting the first direct evidence that PCO1/4 catalyze the O₂-dependent post-translational oxidation of an internal cysteine within intact peptide sequences to produce a sulfinic acid. While some interesting correlations to O₂-sensing pathways are presented, I remain unconvinced that sufficient evidence is presented to make this conclusion. Overall, the majority of experiments are well considered and presented. However, I do have some concerns regarding the presentation of the chemistry and over-interpretation of single turnover experiments which I have described below. Some minor issues are presented as well, but these should be considered stylistic suggestions to improve the clarity of the report.

Major concerns.

1. By definition, cysteine dioxygenase chemistry is a 4 electron process; oxidation of the cysteine-thiol sulfur atom (-II) to the sulfinic acid (+II) moiety. Concomitant with this half-reaction, molecular oxygen accepts these 4 electrons to be reduced to the level of water. Therefore, the frequent digression to present a potential sulfonic acid product (-CysO₃-) [paragraph 5, line 126] is very distracting and takes away from the authors central message. I realize the authors are attempting to explain the formation of the sulfonic acid minority species in MALDI experiments, but the presentation of this material is distracting and potentially misleading. I would recommend spending some time clarifying these discussions.

2. I am not convinced that demonstration of O₂-dependent cysteine oxidation by single turnover is evidence that PCOs serve as an O₂-sensor similar to HIF. For example, the mammalian and bacterial cysteine dioxygenase (CDO) enzymes exhibit a submicromolar apparent K_M-value for oxygen. In fact, in steady-state experiments, the observed k_{cat} remain fixed down to oxygen concentrations nearly equivalent to the enzyme concentration. Therefore, CDO enzymes would make for a poor sensor since any O₂ present would immediately results in product formation.¹ By contrast, clear O₂-saturation kinetics can be observed for the non-heme mononuclear iron HIF or the extensively characterized [FeS]-dependent FNR O₂-sensor.²⁻³ I would argue that in order to justify that PCOs represent a new paradigm for O₂-sensing, the authors should first confirm that O₂-saturation behavior is observed. Otherwise, the single turnover experiments simply indicate a post-translational modification of a target sequences containing cysteine.

Minor issues.

1. I would like to see some characterization of the PCO enzymes. For instance, do these enzyme contain 1 mol of Fe per protein as other CDO enzymes?

2. Experimental conditions omit various key details relevant to the state of the enzymes they purify. For instance, "the coding sequence of Arabidopsis ATE1 was cloned according to gene annotations at TAIR (...) from cDNA. The sequence was flanked by an N-terminal tobacco etch virus (TEV) recognition sequence for facilitated downstream purification..."...The authors never specify what fusion tag they are cutting off with TEV

protease. Ni-IMAC purification is utilized to purify Fe-dependent (presumably) PCO enzymes however, the metal complement is never determined following purification and no exogenous Fe is added during steady-state assays.

1. Crowell, J. K.; Li, W.; Pierce, B. S., Oxidative Uncoupling in Cysteine Dioxygenase Is Gated by a Proton-Sensitive Intermediate. *Biochemistry* 2014, 53 (48), 7541-7548.
2. Sutton, V. R.; Mettert, E. L.; Beinert, H.; Kiley, P. J., Kinetic Analysis of the Oxidative Conversion of the [4Fe-4S]₂⁺ Cluster of FNR to a [2Fe-2S]₂⁺ Cluster. *Journal of Bacteriology* 2004, 186 (23), 8018-8025.
3. Taabazuing, C. Y.; Hangasky, J. A.; Knapp, M. J., Oxygen sensing strategies in mammals and bacteria. *Journal of Inorganic Biochemistry* 2014, 133, 63-72.

Reviewer #2 (Remarks to the Author):

The manuscript "Plant Cysteine Oxidases are Dioxygenases that Directly Enable Arginyl Transferase-Catalyzed Arginylation of N-End Rule Targets" by Mark D. White et al. elucidates the mechanism of amino-terminal Cys oxidation by a recently discovered class of plant enzymes.

The manuscript provides a thorough and complete description of a previously poorly characterized enzyme reaction and shows that it can directly couple to a reaction that has been studied for a long time, amino-terminal arginylation.

The reviewer cannot comment on details of the chemistry involved in peptide synthesis or on NMR analysis.

Regarding posttranslational arginylation, the authors should quote a recent publication by the Varshavsky lab (Wadas et al., 2016, *J Biol Chem.* 2016 Aug 10. pii: jbc.M116.747956 PMID 27510035) that casts doubt on published results of arginylation of amino groups not at the N-terminus (line 319 ff).

Reviewer #3 (Remarks to the Author):

The manuscript by White et al describes studies that investigate Plant Cysteine oxidases (PCO)-mediated cysteinyl oxidation of an ETHYLENE-RESPONSE FACTORS group VII (ERF-VII) -derived peptides and the subsequent arginylation by recombinant arginyl transferase (ATE1), two crucial well-documented biochemical steps toward the recognition and subsequent degradation of N-terminal cysteine-containing protein substrates by the N-end rule pathway. Using mass spectrometry, ¹H NMR spectroscopy and in vitro biochemical assays, the authors demonstrate that PCOs (namely PCO1&PCO4) catalyze the oxidation of N-terminal Cysteine of the synthetic peptide RAP22-11 to form Cys-sulfinic acid using molecular oxygen as co-substrate.

However, the identification, functional characterization and physiological significance of PCOs, including the PCO-mediated cysteinyl oxidation of an ERF-VII-derived peptide to generate Cys-sulfinic acid using molecular oxygen as co-substrate, has been previously demonstrated recently (Daan Weits et al. Nat. Comm. 2014). Albeit, this report does an excellent characterization of the exact nature of the chemical intermediates of the reaction which was not done in the previous report.

In addition the authors confirmed that PCO-catalyzed Cys-oxidation to Cys-sulfinic acid renders a RAP2 peptide capable of subsequent modification by ATE1. However, the cascade of Cys-oxidation to Cys-sulfinic acid and the subsequent arginylation via ATE1 has been well-documented previously in plants, mammalian systems and in vitro. (Hu et al., Nature 2005; Lee et al., PNAS 2005; Daan Weits et al. Nat. Comm. 2014; B Wadas et al. JBC 2016; Davydov and Varshavsky JBC 2000; Manahan and App, Plant Physiol, 1973 ; Hu et al., JBC 2006 ; Garzon et al. FEBS Letters 2007; Licausi et al. Nature 2011; Kwon et al. Science 2002). Of note, the authors have not examined any new physiological inputs and/or biochemical factors that influence Cys-oxidation and the subsequent arginylation by ATE, yet rather confirmed previous findings.

Overall, the current manuscript, though scientifically-sound, does not provide an apparent significant conceptual advance in the field.

Specific points:

1- The finding that PCOs can mediate the oxidation of N-terminal cysteine residues to CSA using molecular oxygen as a co-substrate has been elegantly demonstrated previously (Daan Weits et al. Nat. Comm. 2014). Given that a number of PCOs has been identified in the plant *Arabidopsis thaliana* (5 members). It has been shown that PCO1 & 2 are the most expressed PCO genes and that the modulation of their expression has been shown to exhibit a physiological impact. Why the authors investigated PCO1 & 4, why not PCO1, 2 and 4, or all the five members?; Is there any kinetic or selectivity differences between PCO1/2/4-mediated N-terminal cysteine residues oxidation (particularly in response to different O₂ levels)?; Could PCO 1/4 mediate Cys-oxidation of internal Cys-residues? Can the kinetics of PCO1/2/4-mediated Cys-oxidation be characterized at different levels of O₂?

2- PCOs show sequence homology to Fe(II)-dependent Cysteine dioxygenases (CDO) family of enzymes and since significant biochemical/computational and structural models have proposed some critical motifs/sites for the catalysis of CDO like the 3His ligand system that has been postulated to be necessary for optimal dioxygenation of cysteine as well as the presence of crucial motif of cysteinyltyrosine (Tyr157-Cys93) post-translational modification near the active site that may influence the PCO-mediated Cys-oxidation. (Aluri and de Visser JACS 2007; de Visser and Straganz J Phys Chem 2009; Kumar et al. 2011; Joseph CA Chem Comm 2007 ; Ye et al JBC 2007 ; Simmons CR et al. JBC 2006). To dissect the molecular basis and requirements of PCO-mediated Cys-oxidation, the efficiency of wild-type PCOs versus mutant versions of PCOs (mutations of the aforementioned crucial motifs/sites need to be generated and studied for their role) in mediating Cys-oxidation and its rate should be studied experimentally to faithfully clarify the molecular basis of PCOs action.

3- Given that various physiological inputs (e.g. normoxia, hypoxia and Anoxia, NO-levels, Light conditions, air humidity, etc) can affect ERF-VII transcription factors stabilities (Protein substrates that contain Met-CYS-N-termini) and hence affect the physiological response (Gibbs et al. Mol cell. 2014; Weits et al. Nat. Comm. 2014; Verena et al. Plant Physiol 2016 ; Abaas et al. Curr. Biol. 2015). The current manuscript focuses on one parameter (O₂

deficient or normoxia) on Cys-oxidation of ERF-VII transcription factors-derived peptides that have been previously demonstrated. What are the effects of the following physiological inputs (normoxia, hypoxia and Anoxia, NO-levels, Light conditions, air humidity, ... etc) on PCO-mediated Cys-oxidation to faithfully advance our understanding regarding the physiological implications of N-end rule pathway in plants.

4- While the reviewer appreciate the well developed and controlled experiments to evaluate arginylation, via recombinant ATE1-derived from *Arabidopsis thaliana*, of synthetic peptides bearing either Nt- Cys , Asp , Gly or Cys-sulphonic acid. Others have been much more extensive on investigating both the first and second position effects on mammalian ATE1 substrate recognition (Wadas et al 2016 JBC). A more complete investigation of ATE1-derived from *Arabidopsis thaliana* would be beneficial to obtain insights about plant ATE1 substrate recognition.

5- It is somewhat surprising that key research reports, including the early report about identification of plant Ate1 and arginylation in plants and the recent report that analyze extensively N-terminal arginylation have not been cited by the authors, given that ATE1 and arginylation lie at the heart of this manuscript. (Yoshida et al 2002 plant journal ; Graciet et al 2010 plant journal; Manahan and App, Plant Physiol, 1973 Wadas et al 2016 JBC). In addition the authors cite that the first non Cys-branch Ate1 substrates were only recently identified in mammals (Cha-Molstad 2015, Nat Cell Biol). Prior to this paper has been a number of publications by A. Kashina's group on arginylation targets and the reports on the Ate1 dependent degradation of the caspase generated fragment of BRCA1 by two independent groups (Piatkov et al 2012 PNAS; Xu et al 2012 JBC)

In summary:

Efforts to link the chemical/genetic modulations of PCOs and ATE activity to novel physiological responses and the full biochemical characterization of molecular basis of ATE1/2-mediated arginylation of plant N-end rule substrates in the light of aforementioned comments would improve the scope of current work.

We have listed all relevant and in the following discussed references in a bibliography at the end of our comments to Reviewer 3.

Reviewer 1

Overall this is an interesting paper presenting the first direct evidence that PCO1/4 catalyze the O₂-dependent post-translational oxidation of an internal cysteine within intact peptide sequences to produce a sulfinic acid. While some interesting correlations to O₂-sensing pathways are presented, I remain unconvinced that sufficient evidence is presented to make this conclusion.

Response: We thank the Reviewer for her/his positive comments and the effort and time spent for refereeing. We have commented below in detail our analysis of 'N-terminal' (rather than 'internal') Cys residues to deliver the first molecular evidence for the existing link between enzymatic Cys oxidation in the presence of molecular O₂ by Plant Cysteine Oxidases (PCOs) and N-terminal arginylation. This mechanism was long speculated about but not yet demonstrated. In addition, the chemical identity of the modified Cys residue remained obscure to date and here we have characterized its actual chemical modification for the first time. The aim of the work was not to describe or characterise the oxygen sensing mechanism; their oxygen sensitivity was already published in *Nature Communications* in 2014 (Weits et al., Plant cysteine oxidases control the oxygen-dependent branch of the N-end-rule pathway, *Nature Comms* 5, 3425) and detailed kinetic characterization of PCO activity was not the intention of this report. We reference the oxygen sensitivity assays conducted in the Weits et al. paper and adjust the text of our manuscript to reiterate the aims of our work (see below for details (Weits et al., 2014)).

Overall, the majority of experiments are well considered and presented. However, I do have some concerns regarding the presentation of the chemistry and over-interpretation of single turnover experiments which I have described below. Some minor issues are presented as well, but these should be considered stylistic suggestions to improve the clarity of the report.

Response: We went through the text carefully to make sure that we very clearly describe our intentions and do not infer that we are investigating the oxygen sensitivity in this manuscript, but do refer to Weits et al. (*Nature Comms*, 2014) and the oxygen sensitivity work there. Furthermore we make it clear that we are performing end-point assays (in order to characterize products of the reaction) and not single turnover experiments.

Major points:

1. By definition, cysteine dioxygenase chemistry is a 4 electron process; oxidation of the cysteine-thiol sulfur atom (-II) to the sulfinic acid (+II) moiety. Concomitant with this half-reaction, molecular oxygen accepts these 4 electrons to be reduced to the level of water. Therefore, the frequent digression to present a potential sulfonic acid product (-CysO₃-) [paragraph 5, line 126] is very distracting and takes away from the authors central message. I realize the authors are attempting to explain the formation of the sulfonic acid minority species in MALDI experiments, but the presentation of this material is distracting and potentially misleading. I would recommend spending some time clarifying these discussions.

Response: We thank the Reviewer for pointing at this part which requires further clarification. We absolutely agree with the Reviewer that the oxidation of a cysteine thiol by oxygen should result in formation of Cys-sulfinic acid, however given that Cys-sulfinic acid and Cys-sulfonic acid have been extensively discussed in the N-end rule literature as potential yet unknown Cys oxidation products

that both enable subsequent arginylation (also directly demonstrated for the first time *in this manuscript* (Figure 4a); (Varshavsky, 2011; Tasaki, T *et al.*, 2012; Gibbs *et al.*, 2014)), we wished to remain open minded to the possibility that further PCO-mediated oxidation of Cys-sulfinic acid to Cys-sulfonic acid could take place (possibly with O₂ or H₂O as a co-substrate). The Reviewer rightly alerted us to the fact that our discussion of Cys-sulfonic acid as a potential product was distracting. However, we considered it important to strike a balance between maintaining the clarity of our text and conveying the important message (especially to the N-end rule community) that we investigated both species as possible products, but flagged Cys-sulfonic acid as artefact. We therefore made the following modifications to the text and Supplementary Figure 1:

A, Supplementary Figure 1 (referred to in main text page 4, lines 91/92) has been modified to remove PCO from the scheme. We intended that this scheme represents the potential Cys-thiol oxidations according to N-end rule literature, but describing them at this stage as *potential* PCO-catalysed reactions was misleading. In particular, including formation of Cys-sulfonic acid as a potential PCO-catalysed reaction in such a scheme could easily imply that this was our eventual finding. We have removed PCOs from the scheme and altered the legend to more accurately convey that these Cys oxidations are those reported to precede arginylation in the Arg/Cys branch of the N-end rule.

B, Main text page 5, lines 129/130: we have removed 'rather than the fully oxidised Cys-sulfonic acid (CysO₃) (Supplementary Figure 1)'.

C, Main text page 7, lines 166-170: we have inserted 'Although homology between the PCOs and CDOs leads us to the predisposition that they will perform similar chemistry (i.e. catalyse Cys-sulfinic acid formation), both Cys-sulfinic and Cys-sulfonic acid are proposed to be Arg transferase substrates in the Arg/Cys branch of N-end rule mediated protein degradation therefore both were initially considered as potential products of the PCO-catalysed reaction.'

2. I am not convinced that demonstration of O₂-dependent cysteine oxidation by single turnover is evidence that PCOs serve as an O₂-sensor similar to HIF. For example, the mammalian and bacterial cysteine dioxygenase (CDO) enzymes exhibit a submicromolar apparent K_M-value for oxygen. In fact, in steady-state experiments, the observed k_{cat} remain fixed down to oxygen concentrations nearly equivalent to the enzyme concentration. Therefore, CDO enzymes would make for a poor sensor since any O₂ present would immediately results in product formation.¹ By contrast, clear O₂-saturation kinetics can be observed for the non-heme mononuclear iron HIF or the extensively characterized [FeS]-dependent FNR O₂-sensor.²⁻³ I would argue that in order to justify that PCOs represent a new paradigm for O₂-sensing, the authors should first confirm that O₂-saturation behavior is observed. Otherwise, the single turnover experiments simply indicate a post-translational modification of a target sequences containing cysteine.

Response: We would like to respectfully highlight that the purpose of our manuscript is not to demonstrate that the PCOs represent a new paradigm in O₂-sensing. The O₂-sensing potential of the PCOs was published by Weits *et al.* in *Nature Communications* in 2014 (subsequent detailed kinetic studies of the PCOs are ongoing and will be published at a later date). Rather the purpose of our paper is to provide the molecular validation of the proposed underlying chemistry of the PCO- and ATE-catalysed reactions. Besides our work, there are only two publications available, where N-terminal arginylation (following presumed Cys oxidation) is really shown by a 'molecular method', that is arginylation of mammalian RGS4 by mass spectrometry (Kwon, Y *et al.*, 2002; Hu, R *et al.*, 2005); further, these reports do not adequately address the identity of modification of the second residue, formerly a Cys, nor does it confirm the N-terminal location of the peptide modifications. The evidence presented in our work importantly addresses such molecular assumptions to date absent from this field, e.g. demonstrates that molecular O₂ is directly incorporated into the product of the PCO-catalysed (end-point) reaction, confirms (by MSMS and NMR) the chemical nature of

the product and provides the first direct evidence for arginylation of an N-terminal Cys-sulfinic acid residue. This level of molecular characterisation has not yet been reported for the Arg/Cys branch of the N-end rule pathway and we feel represents an important advance in the field in and of itself.

While the detailed O₂-sensing kinetic properties of the PCOs will be of significant interest (as stated in the main text page 13, lines 313-316), particularly in comparison to mammalian and bacterial O₂-sensing strategies, we strongly feel that presentation of this data is not within the scope of this particular manuscript, rather we seek to provide a chemically robust platform for subsequent studies of this pathway. Although not rigorously determined using steady state kinetic techniques, a level of O₂ sensitivity has already been reported for the PCOs in the original publication describing their biological function (Weits *et al.*, 2014).

In order to make it clearer in the manuscript that we are reporting molecular validation of steps in the Arg/Cys branch of the N-end rule pathway (including validation of the role of O₂ as a cosubstrate) and not trying to define the PCOs as oxygen *sensors* at this stage, we have made the following changes to the text:

A, Page 5, lines 123/124: We have inserted 'Validation of the chemical steps in the Arg/Cys branch of the N-end rule pathway is currently limited, both in animals and plants.'

B, Page 8, lines 173/174: We have inserted '...and thus confirm a direct connection between molecular O₂ and PCO activity...'

B, Page 13, lines 311/312: We have replaced 'Their direct use of O₂ supports the proposal that these enzymes are plant O₂ sensors' with 'Their direct use of O₂ supports the proposal that these enzymes *may* act as plant O₂ sensors'.

Minor Points:

1. I would like to see some characterization of the PCO enzymes. For instance, do these enzyme contain 1 mol of Fe per protein as other CDO enzymes?

Response: The Reviewer raises a good point in that it is important to demonstrate some initial characterisation of PCO1 and PCO4. Importantly we should demonstrate that the enzymes contain Fe, as reported for their CDO homologs (Weits *et al.*, 2014), especially considering that we report that they catalyse similar reactions (cysteine/cysteinylation). To this end, we have conducted additional experiments, the results of which we have added to Supplementary Figure 2. We report:

A, Multiple angle light scattering experiments which confirm that the purified enzymes are monomeric.

B, Micro-PIXE (Particle Induced X-ray Emission) experiments which detect and quantify elemental composition of protein samples (Garman & Grime, 2005). These results revealed Fe content in both PCO1 and PCO4 at a ratio of ~0.3 Fe atoms per monomer. This is the level of Fe present in recombinant protein as purified following expression in *E.coli*, i.e. without supplementary iron, and is in line with the levels of Fe co-purified with recombinant CDOs (where levels vary from 0.1-0.7 Fe atoms per monomer (Imsand *et al.*, 2012), as well as with the human oxygen sensor PHD2 (McNeill *et al.*, 2005). The microPIXE results also revealed notable quantities of nickel, however this is likely due to leaching of Ni²⁺ with enzyme elution during their purification by Ni²⁺-affinity chromatography.

C, Continuous wave EPR spectra that do not demonstrate any significant enzyme-associated resonance, suggesting that the iron content revealed by micro-PIXE is in the EPR-silent Fe(II) state

rather than the EPR-‘active’ Fe(III) state.

This data is collectively referred to in the main text on page 7, lines 149-152 (and referred to briefly again in the Discussion page 14 lines 320-321). We have deliberately limited our discussion of these results in the main text to a single sentence so as not to distract from the main narrative of our manuscript, which is the characterisation of the products formed by these enzymes, and thus molecular validation of the Arg/Cys branch of the N-end rule pathway. The reader is nevertheless referred to Supplementary Figure 2c-e so those with an interest in this data can examine the evidence.

2. Experimental conditions omit various key details relevant to the state of the enzymes they purify. For instance, "the coding sequence of Arabidopsis ATE1 was cloned according to gene annotations at TAIR (...) from cDNA. The sequence was flanked by an N-terminal tobacco etch virus (TEV) recognition sequence for facilitated downstream purification..."...The authors never specify what fusion tag they are cutting off with TEV protease. Ni-IMAC purification is utilized to purify Fe-dependent (presumably) PCO enzymes however, the metal complement is never determined following purification and no exogenous Fe is added during steady-state assays.

Response: We apologize for having omitted some details on the use of our recombinant ATE1 protein in the previous version of the manuscript. Actually, we have NOT cleaved the fusion protein and did therefore not make use of the TEV recognition site. We added this information to the Methods section to keep transparency about the procedures and plasmid material used. We have used a Gateway-compatible expression vector (pDEST17) from Invitrogen and had not further described this in the Methods part of the paper. pDEST17 contains a hexahistidine encoding sequence 5' of the Gateway recombination site and by adding the TEV recognition site in the forward cloning primer, we had the opportunity to get a tag-free ATE1 as well. However, this was not necessary because the His6-ATE1 has a strong and specific arginylation activity.

We wrote on p.11 l. 250 f. "To this end, we produced recombinant hexahistidine tagged Arabidopsis ATE1 (Supplementary Figure 5)" and on p. 20 l. 469-470 ff. „cloning into pDONR201 (Invitrogen) followed by an LR reaction into the vector pDEST17 (Invitrogen). The N-terminal hexahistidine fusion was expressed...". So the information was basically already in the submitted version but it remained unclear that we did not cleave the His6 tag off. We checked and edited the respective parts in the results and methods sections.

Concerning PCOs, as discussed above, we have added data determining the quantity and oxidation state of the iron that co-purifies with the PCO enzymes. We also note that in contrast to the CDOs (Imsand *et al.*, 2012), the enzymes did not need to be supplemented with Fe or ascorbate for activity during the end-point assays which we conducted in this study. This has been included in the main text (p. 7, l. 162 f.) and in the Methods (p. 21, l. 497), as it may reflect differences in reaction parameters/behaviour between the two classes of enzymes.

Reviewer 2

The manuscript "Plant Cysteine Oxidases are Dioxygenases that Directly Enable Arginyl Transferase-Catalyzed Arginylation of N-End Rule Targets" by Mark D. White et al. elucidates the mechanism of amino-terminal Cys oxidation by a recently discovered class of plant enzymes.

Response: We thank the Reviewer for the positive comments and acknowledging the first characterization of this enzymatic mechanism.

The manuscript provides a thorough and complete description of a previously poorly characterized enzyme reaction and shows that it can directly couple to a reaction that has been studied for a long time, amino-terminal arginylation.

Response: We were very happy to see the positive perception of our work by the Reviewer and acknowledging that the described enzymatic reactions, including the long known and debated arginylation, was underexplored and remained unclear since its discovery.

The reviewer cannot comment on details of the chemistry involved in peptide synthesis or on NMR analysis.

Response: Standard techniques were used in this context.

Regarding posttranslational arginylation, the authors should quote a recent publication by the Varshavsky lab (Wadas et al., 2016, J Biol Chem. 2016 Aug 10. pii: jbc.M116.747956 PMID 27510035) that casts doubt on published results of arginylation of amino groups not at the N-terminus (line 319 ff).

Response: We thank this Reviewer for their positive comments about the manuscript. Their suggested citation of the recent publication by Wadas et al (*J Biol Chem*, 2016 291: 20976) describes experiments that address questions surrounding the behaviour of different mammalian ATE1 isoforms, particularly towards non-N-terminal but rather internal amino acid residues. We have cited this publication at the relevant points in the Discussion (please see page 15 lines 348-351). This work also describes ¹⁴C Arg incorporation assays showing mouse ATE1 mediated Arg incorporation into a Cys-sulfonic acid N-terminus but not an unoxidised free Cys N-terminus, and we have therefore referenced this publication in the current version of the manuscript as relevant (page 11, line 259-260; page 12, line 285). Interestingly, it was not possible to test whether Cys sulfinic acid may act as potential arginylation acceptor since the required Fmoc-labelled amino acids for peptide synthesis are not available and difficult – if at all possible – to synthesize chemically.

Reviewer 3

The manuscript by White et al describes studies that investigate Plant Cysteine oxidases (PCO)-mediated cysteinyl oxidation of an ETHYLENE-RESPONSE FACTORS group VII (ERF-VII) –derived peptides and the subsequent arginylation by recombinant arginyl transferase (ATE1), two crucial well-documented biochemical steps toward the recognition and subsequent degradation of N-terminal cysteine-containing protein substrates by the N-end rule pathway.

Response: We thank the Reviewer for her/his effort in assessing our manuscript and the comments and suggestions made. We hope to improve the manuscript by further including better descriptions on the rationale of our experiments and more details on the actual state-of-the-art as this might have fallen too short in our introduction.

Our main point is that we wish to deliver for the first time detailed molecular evidence validating a “missing link” in the field of N-end rule research. We define and connect for the first time enzyme-catalysed oxidation of an N-terminal Cys residue to Cys-sulfinic acid with N-terminal arginylation. By this, we deliver novel mechanistic insights into targeted proteolysis with clearly shown impacts in the response of organisms to the environment ((Gibbs *et al.*, 2011; Licausi *et al.*, 2011; Weits *et al.*, 2014).).

In our opinion, one of the main ‘uncertainties’ in the field of targeted proteolysis via the N-end rule pathway actually concerns the current ‘state-of-the-art’, that is the real evidence-based characterization of molecular events researchers have documented by their data. Several molecular steps within the enzymatic N-end rule reaction cascade leading to posttranslational modifications of target proteins especially – but not exclusively – at their N-termini, are still obscure to date, and none have been rigorously chemically validated (at least for the Arg/Cys branch). This is especially true for the plant field where important biological functions of the N-end rule pathway were described but the molecular basis largely lacks to date. Detailed validation of both Cysteine oxidation and consequent arginylation is delivered now for the first time in our work.

We would respectfully refute that cysteine oxidation and arginylation are ‘well-documented’ as Reviewer 3 writes, either in plants or more widely (at least for Cys oxidation) in other organisms. In plants in particular, the identification and biological role for these post-translational modifications is just emerging yet has predominantly been based on genetic observations (Gibbs *et al.*, 2011; Licausi *et al.*, 2011; Weits *et al.*, 2014).

Arginylation as a posttranslational modification is known since 1963 (Kaji *et al.*, 1963), an aminoacyl transferase function in plants (rice and wheat) since 1973 ((Manahan & App, 1973); no relation to the amino terminus though) and its involvement in the N-end rule pathway has been speculated since 1988 (by both Ciechanover *et al.* and Bohley *et al.* (Bohley *et al.*, 1988; Ciechanover *et al.*, 1988)). However, the only experimental evidence for arginylation occurring at the N-terminus of a protein was published for mouse BRCA1 (N-terminal Aspartate (Piatkov *et al.*, 2012)), the acylamino-acid-releasing enzyme PpAARE from moss (N-terminal Aspartate (Hoernstein *et al.*, 2016)) and for mouse RGS4/5 (regulator of G protein signaling), where Cys oxidation prior to arginylation has also been reported ((Davydov & Varshavsky, 2000), see below). Only arginylation of BRCA1 and RGS4/5 result in destabilization according to the N-end rule pathway.

Moreover, so far, RGS4/5 are the only two cases known where N-terminal Cys are reported to be oxidized (to Cys-sulfonic acid), and subsequently serve as arginylation substrates of ATEs, leading to degradation by the N-end rule pathway (Lee *et al.*, 2005). This is also the only example in the literature where fragmentary mass spectrometry is used to indisputably demonstrate arginylation

of a RGS4 peptide with a +48 Da mass consistent with the presence of an oxidized Cysteine (Cys sulfonic acid) at the +2 position (Kwon, Y *et al.*, 2002). We would note that this is not coupled with NMR-confirmation of structure as presented in our work, nor is there any demonstration of incorporation of molecular O₂, though an enzymatic process is suggested and stability of RGS4/5 is enhanced in hypoxia (Lee *et al.*, 2005). Nitric oxide (NO) has also been reported to promote Cys-sulfonic acid formation and subsequent arginylation at the N-terminus of RGS4/5 in vivo and in vitro, via nitrosylation followed by oxidation, however the mechanism was not discussed nor any molecular evidence of nitrosylation and/or subsequent oxidation provided (Hu, R *et al.*, 2005). Nevertheless this proposal supports biological data, e.g. stabilization of Nt-Met-Cys-substrates in cells lacking NO synthases.

The data from RGS4/5 therefore suggest, with some chemical validation, that Cys-sulfonic acid is a substrate for ATE-mediated arginylation. Cys-sulfinic acid is consistently proposed as a potential substrate for ATE-mediated arginylation, yet there is no evidence for this in the literature and this 'dogma' appears to have arisen solely due to the similarity in structure of Cys-sulfinic acid to Asp and Glu which are also ATE substrates (Kwon, Y *et al.*, 2002). While Weits *et al.* go some way to suggest O₂-dependent PCO-catalysed formation of Cys-sulfinic acid, their evidence is by no means indisputable (Weits *et al.*, 2014).

Our work therefore significantly advances the field by unequivocally demonstrating:

1. PCO-catalysed oxidation of Nt-Cys-RAP2 to form Cys-sulfinic acid, incorporating molecular O₂ directly into the product. Please note that by characterizing the first plant cysteinyl oxygenases, the work is also a significant advance in the field of oxygenase enzymology.
2. Validation of specific activity of the first recombinantly produced plant ATE1 enzyme towards an Arg/Cys branch N-end rule proposed substrate.
3. The first demonstration that Cys-sulfinic acid is a substrate for ATE-mediated arginylation, with molecular validation of this reaction by LC-MS and MS/MS.

This reviewers comments highlight to us that we have not been assertive enough in describing the advances made by our work in the main text of the manuscript. We have therefore made the following modifications:

1. Abstract modified in lines 57-59 to read: 'This is the first molecular evidence showing N-terminal Cys-sulfinic acid formation and arginylation by N-end rule pathway components, and the first ATE1 substrate in plants'
2. Added to page 5, lines 135-137: 'This provides the first evidence that Nt-Cys-sulfinic acid is a *bona fide* substrate for N-end rule mediated arginylation.'
3. Added to page 11, line 243 onwards: 'Cys-sulfinic acid has been proposed as a substrate for ATE1 on the basis of its structural homology with known ATE1 substrates Asp and Glu, but evidence has only been reported to date for arginylation of Cys-sulfonic acid.'
4. Added to page 14, line 327 onwards: 'While both Cys-sulfinic and Cys-sulfonic acid are repeatedly reported as potential arginylation substrates(Varshavsky, 2011; Tasaki, T. *et al.*, 2012; Gibbs *et al.*, 2014), detailed evidence has only been presented to date for arginylation of Cys-sulfonic acid(Kwon, Y *et al.*, 2002; Hu, R *et al.*, 2005) and this only in a mammalian system.'

Using mass spectrometry, 1H NMR spectroscopy and in vitro biochemical assays, the authors demonstrate that PCOs (namely PCO1&PCO4) catalyze the oxidation of N-terminal Cysteine of the synthetic peptide RAP22-11 to form Cys-sulfinic acid using molecular oxygen as co-substrate. However, the identification, functional characterization and physiological significance of PCOs, including the PCO-mediated cysteinyl oxidation of an ERF-VII-derived peptide to generate Cys-sulfinic acid using molecular oxygen as co-substrate, has been previously demonstrated recently (Daan Weits et al. Nat. Comm. 2014). Albeit, this report does an excellent characterization of the exact nature of the chemical intermediates of the reaction which was not done in the previous report.

Response: Following from our comments above, we thank the Reviewer for acknowledging our detailed work on molecularly characterizing the true and still obscure oxidation state of the 'oxidized cysteine'. However, we disagree that in the mentioned work (Weits *et al.*, 2014) it was indeed demonstrated that 'PCO-mediated cysteinyl oxidation of an ERF-VII-derived peptide to generate Cys-sulfinic acid'.

We fully acknowledge that the PCOs were identified and their biological and physiological function characterised by Weits *et al.* (though there is a strong argument that one publication does not constitute well documented evidence). However, it is very important to note that the approach used in this report to characterise the reaction is not sufficient to define the PCOs as cysteinyl dioxygenases (this is readily acknowledged by the authors of the paper (personal communication Francesco Licausi). Although Weits *et al.* demonstrated consumption of molecular oxygen, this is not evidence that molecular oxygen is a co-substrate, particularly when additional factors Fe(II) and ascorbate were also present in the reaction mix (oxygen can be consumed upon oxidation of Fe-ascorbate complexes (Hamed *et al.*, 1988; Dorey *et al.*, 1993)). Further, HPLC-elution characteristics similar to those of CSA do not verify in themselves that the product is CSA itself. Other potential products (e.g. Cys-sulfonic acid) were not tested for their HPLC elution characteristics by Weits *et al.* (personal communication, Francesco Licausi).

We would like to emphasise that the detailed chemical characterisation of an enzyme-catalysed reaction, as demonstrated in our manuscript, is particularly important when studying a newly identified families of enzymes such as the PCOs and those where only genetic data exist like for ATEs. Both the PCOs and ATEs are of significant potential interest to the hypoxia, agrochemical, enzymology and plant biology communities, with significant potential for modulating their activity for agricultural benefit. As such it is vital that the chemical foundations of further biological or biochemical study are defined, and it is critical that the underpinning chemical identity of the product is confirmed before further studies are built on potentially false assumptions (as validated extensively in the human hypoxia-sensing literature). This is particularly true given the ambiguity in the N-end rule literature regarding Cys-sulfinic vs. Cys-sulfonic acid modifications. We strongly believe therefore that our work is a highly valuable contribution to the field that will be highly cited.

In addition the authors confirmed that PCO-catalyzed Cys-oxidation to Cys-sulfinic acid renders a RAP2 peptide capable of subsequent modification by ATE1. However, the cascade of Cys-oxidation to Cys-sulfinic acid and the subsequent arginylation via ATE1 has been well-documented previously in plants, mammalian systems and in vitro. (Hu et al., Nature 2005; Lee et al., PNAS 2005; Daan Weits et al. Nat. Comm. 2014; B Wadas et al. JBC 2016; Davydov and Varshavsky JBC 2000; Manahan and App, Plant Physiol, 1973 ; Hu et al., JBC 2006 ; Garzon et al. FEBS Letters 2007; Licausi et al. Nature 2011; Kwon et al. Science 2002).

Response: As described above, we would reiterate that there is *no evidence* to date that Cys-

sulfinic acid is a substrate for modification by ATE1, and much of the evidence for Cys-sulfonic acid modification has not been chemically validated. Only in the case of Hu et al 2005 and Kwon et al. (Kwon, Y *et al.*, 2002) are mass spectrometry data presented that define Cys-sulfonic acid as a substrate for arginylation in the mammalian system. While other work has defined Nt Cys-sulfonic as an ATE-substrate, e.g. Wadas et al. (Wadas *et al.*, 2016), this is only shown via antibody-based studies and not the more definitive mass spectrometry. It is worth noting that no one has ever before been able to demonstrate that Cys-sulfinic acid is a substrate for ATE using peptide studies, as Nt-Cys-sulfinic acid would rapidly become oxidised to Cys-sulfonic acid during the peptide synthesis process. This further emphasises the value of our work.

We appreciate that the Reviewer has extensively examined the literature, however we would like to summarise the findings of each of the papers cited in order to support our assertion that our findings are a major step forward in the field. We present these here in chronological order:

(Manahan & App, 1973):

see below at 5); aminoacyl transfer activity in rice and wheat extracts, nature of enzyme and acceptor position unclear, speculation that N-terminus could serve as acceptor (compare Soffer, JBC 1971).

(Davydov & Varshavsky, 2000):

see below at 5); N-terminal initiator methionine of RGS4 (MC-starting protein) replaced with arginine, cause and mechanism unclear, chemical state of Cys2 unclear.

(Kwon, Y *et al.*, 2002):

MS sequencing identified Cys sulfonic acid, suggested an enzymatic oxidation rather than uncatalyzed reaction.

(Hu, R *et al.*, 2005):

nitric oxide (NO) identified as RGS oxidizing agent in vivo, Cys oxidation state unclear (or the discussion again opened, after Kwon et al. Science 2002 suggested Cys sulfonic acid as product), potential role of S-Nitrosylation discussed.

(Lee *et al.*, 2005):

RGS4/5 only two cases known where N-terminal Cys gets oxidized, gets arginylated, and degraded by the N-end rule pathway, Cys oxidation state discussed: Cys sulfinic (cysteinic; CysO₂) or sulfonic (cysteic; CysO₃) acid but remained elusive.

(Hu *et al.*, 2006):

purified ATE1 variants from cell cultures can arginylate N-terminal Glu of reporter protein alpha-lactalbumin, verified by Edman degradation, importantly, GRP78 (BiP) and protein-disulfide isomerase were suggested as putative N-end rule substrates, we have therefore included this information in the manuscript.

(Garzón *et al.*, 2007):

This paper is the first description of candidate E3 ubiquitin ligase PRT6 from Arabidopsis by the Bachmair lab. It contains genetic data on prt6 mutants and compares half-life of artificial reporters to the wild type.

(Licausi *et al.*, 2011):

This paper describes a function of the N-end rule pathway in hypoxia response, oxygen-dependent instability of ERFVII transcription factors and is partially conflicting with the back-to-back paper of Gibbs et al. Nature 2011 regarding flooding tolerance caused by

impaired N-end rule pathway function.

(Weits *et al.*, 2014):

see above and below.

(Wadas *et al.*, 2016):

This in the context of arginyl transfer very important paper was not yet published when we submitted our work. It contains detailed analyses about N-terminal requirement enabling for Ate1 binding determined by peptide arrays. These arrays contained one sequence based on Rgs4 with „N-terminal Cys-sulfonate“ (Cys sulfonic acid) to test for Ate1 binding. Cys sulfonic acid is not amenable for chemical synthesis as outlined here, therefore neither Wadas *et al.* nor we could test it in our peptide-based approaches without the use of a Cys oxidizing enzyme. The latter is extensively described in our paper.

Of note, the authors have not examined any new physiological inputs and/or biochemical factors that influence Cys-oxidation and the subsequent arginylation by ATE, yet rather confirmed previous findings. Overall, the current manuscript, though scientifically-sound, does not provide an apparent significant conceptual advance in the field. Overall, the current manuscript, though scientifically-sound, does not provide an apparent significant conceptual advance in the field.

Response: We reiterate that we believe our paper represents a major advance in the field, as outlined above, as we have shown the molecular function of three members of two plant N-end rule enzyme classes, namely PCO1, PCO4, and ATE1. Not only have we chemically validated an assumed connection between molecular O₂ and PCO activity (through the use of 18-O₂ experiments), we have unambiguously defined the product of the PCO reaction and demonstrated for the first time Nt-Cys-sulfonic acid product is a substrate for ATE-1. We are convinced that this mechanistic insight is important in the light of speculation in the field which is ongoing since 2009 (Graciet *et al.*, 2009) – or even since 1973 (Manahan & App, 1973) about ‘potential Arg transferases in plants’ and the mechanistic causes of degradation of Met-Cys-starting proteins such as ERFVIs. All of these are major advances in a field which has been based on assumption of chemical characteristics based on biological data. Further, the degradation of ERF-VIs and its physiological including the agronomical implications, have been intensively discussed in the literature and on conferences since the two back-to-back papers of Gibbs *et al.* and Licausi *et al.* in Nature in late 2011 (Gibbs *et al.*, 2011; Licausi *et al.*, 2011). We present data bridging the state-of-the-art knowledge, and assert that we do not confirm previous ‘findings’ but rather some of the previously made ‘speculations’ and ‘assumptions’.

We strongly feel that the value of this manuscript lies in these findings which lay the chemical foundations for subsequent studies on these enzymes. As modulation of the activity of these enzymes has the potential to impact on plant development and stress tolerance, the effects of additional biochemical factors that influence Cys oxidation and arginylation are more appropriate in multiple follow-up publications.

We hope that we have convinced the Reviewer of the importance of the work we present in this manuscript. Our responses to the below requests by the Reviewer for further studies reflect these arguments, and while the Reviewer makes excellent suggestions for further investigations of this pathway (for which we thank her/him), we argue in each case that these types of investigations are not appropriate for the current manuscript and instead will form the basis of future reports. Indeed many are already underway.

Specific points:

1- The finding that PCOs can mediate the oxidation of N-terminal cysteine residues to CSA using molecular oxygen as a co-substrate has been elegantly demonstrated previously (Daan Weits *et al.* *Nat. Comm.* 2014). Given that a number of PCOs has been identified in the plant *Arabidopsis thaliana* (5 members). It has been shown that PCO1 & 2 are the most expressed PCO genes and that the modulation of their expression has been shown to exhibit a physiological impact. Why the authors investigated PCO1 & 4, why not PCO1, 2 and 4, or all the five members?; Is there any kinetic or selectivity differences between PCO1/2/4-mediated N-terminal cysteine residues oxidation (particularly in response to different O₂ levels)?; Could PCO 1/4 mediate Cys-oxidation of internal Cys-residues? Can the kinetics of PCO1/2/4-mediated Cys-oxidation be characterized at different levels of O₂?

Response: As described above, we respectfully disagree that there is 'elegant demonstration' of PCO-mediated catalysis using molecular O₂ in Weits *et al.* (Weits *et al.*, 2014). As stated above, they demonstrate that O₂ is consumed, but it is not demonstrated that it is incorporated into the product. Only by showing this incorporation of molecular O₂ into the product is a direct correlation between O₂ availability and the N-end rule pathway made to confirm their ability to act as oxygen sensing enzymes. With regards our choice of PCO isoform to study, sequence comparisons (see (Weits *et al.*, 2014)) suggest that PCO1 and 2 show close sequence homology, as do PCO4 and 5, with PCO3 showing similarities to both. We therefore selected PCO1 and 4 to study in order to 1) verify that an as-yet unstudied enzyme (PCO4) catalysed the same reaction as predicted for PCO1, and 2) that we included enzymes from both 'sides' of the PCO family. Although PCO1 and PCO2 are the most highly expressed of the five isoforms, and induced by hypoxia, this does not preclude important biological roles for the other isoforms. To rationalise the homologs we selected to study we have inserted the text 'representatives of the 2 different PCO 'subclasses' based on sequence identity and expression behavior²⁸' into page 5, line 127 and 'different 'subclasses' of ' into page 13, line 310.

Our paper aimed to molecularly characterise the reaction catalysed by the PCOs in end-point reactions, thus we deemed it sufficient to prove this for two members of the family. We agree with the Reviewer that a study of the kinetic parameters of each member of the family will be of significant interest, particularly with respect to oxygen. However, to do this thoroughly is a major undertaking that does not fall within the scope of this report, and will form the basis of a future manuscript, a point we bring up in the discussion (lines 313 to 316, pages 13 to 14).

Please note that we have not extensively investigated whether the PCOs catalyse oxidation of internal Cys residues. Given that their (primary) biological substrate has been identified as N-terminal Cysteine, and biological data points towards N-terminal Cysteines being oxidised as part of the N-end rule pathway, it was this reaction that we sought to validate. Further reactions of PCOs with alternative substrates will be the subject of future studies, and without alternative biologically-identified substrates it would be very difficult to predict flanking sequence requirements. We would note that Weits *et al.* reported that in their system the PCOs could not oxidize an internal Cysteine residue (Weits *et al.*, 2014).

2- PCOs show sequence homology to Fe(II)-dependent Cysteine dioxygenases (CDO) family of enzymes and since significant biochemical/computational and structural models have proposed some critical motifs/sites for the catalysis of CDO like the 3His ligand system that has been postulated to be necessary for optimal dioxygenation of cysteine as well as the presence of crucial motif of cysteinyltyrosine (Tyr157-Cys93) post-translational modification near the active site that may influence the PCO-mediated Cys-oxidation. (Aluri and de Visser *JACS* 2007; de Visser and Straganz *J Phys Chem* 2009; Kumar *et al.* 2011; Joseph *CA Chem Comm* 2007 ; Ye *et al JBC* 2007 ;

Simmons CR et al. JBC 2006). To dissect the molecular basis and requirements of PCO-mediated Cys-oxidation, the efficiency of wild-type PCOs versus mutant versions of PCOs (mutations of the aforementioned crucial motifs/ sites need to be generated and studied for their role) in mediating Cys-oxidation and its rate should be studied experimentally to faithfully clarify the molecular basis of PCOs action.

Response: The Reviewer is right that it would be of great interest to dissect the molecular strategy by which the PCOs catalyse Cys oxidation and it could be informative to utilize the vast body of work conducted on the CDO enzymes (whose mechanism has still not been fully elucidated) to guide a mutagenesis study. Once again however, we feel that this is for a future study and that in order to satisfactorily dissect a mechanism, there first needs to be chemical validation of the catalyzed reaction. That is what we present in this manuscript.

3- Given that various physiological inputs (e.g. normoxia, hypoxia and Anoxia, NO-levels, Light conditions, air humidity, etc) can affect ERF-VII transcription factors stabilities (Protein substrates that contain Met-CYS-N-termini) and hence affect the physiological response (Gibbs et al. Mol cell. 2014; Weits et al . Nat. Comm.2014; Verena et al . Plant Physiol 2016 ; Abaas et al. Curr. Biol. 2015). The current manuscript focuses on one parameter (O2 deficient or normoxia) on Cys-oxidation of ERF-VII transcription factors-derived peptides that have been previously demonstrated. What are the effects of the following physiological inputs (normoxia, hypoxia and Anoxia, NO-levels, Light conditions, air humidity, ... etc) on PCO-mediated Cys-oxidation to faithfully advance our understanding regarding the physiological implications of N-end rule pathway in plants.

Response: Our work does not aim at novel physiological insights as there are other specialist groups in the N-end rule and dioxygenase community addressing this. Our study has a precise focus on 1) molecular enzymology of PCO1/4; 2) addressing speculations and uncertainties about the molecular link between PCO/Cys oxidation/ATE1; 3) demonstrating Arg transfer function of ATE1 for the first time. We agree that we could investigate the effect of various factors on PCO activity, but the point of the PCO work in this manuscript is to provide the chemical validation of the reaction that these enzymes catalyse. This is a necessary foundation from which the effects of different physiological inputs can be determined. (We also suggest that it is likely that a number of the physiological factors which impact on ERFVII stability, e.g. light conditions, humidity, do so by directly or indirectly influencing other components in its regulation and not PCO activity).

4- While the reviewer appreciate the well developed and controlled experiments to evaluate arginylaton, via recombinant ATE1-derived from Arabidopsis thaliana, of synthetic peptides bearing either Nt- Cys , Asp , Gly or Cys-sulphonic acid. Others have been much more extensive on investigating both the first and second position effects on mammalian ATE1 substrate recognition (Wadas et al 2016 JBC). A more complete investigation of ATE1-derived from Arabidopsis thaliana would be beneficial to obtain insights about plant ATE1 substrate recognition.

Response: We fully agree with the Reviewer and find a detailed analysis of ATE1 binding characteristics and enzymology important. However, we feel that this work is far beyond the scope of our paper which does not aim at a full description of plant ATE1 function such as Wadas et al. (Wadas et al., 2016) for mammalian Ate1. Moreover, Wadas et al. was unpublished at the time of submission.

5- It is somewhat surprising that key research reports, including the early report about identification

of plant Ate1 and arginylation in plants and the recent report that analyze extensively N-terminal arginylation have not been cited by the authors, given that ATE1 and arginylation lie at the heart of this manuscript. (Yoshida et al 2002 plant journal ; Graciet et al 2010 plant journal; Manahan and App, Plant Physiol, 1973 Wadas et al 2016 JBC).

Response:

(Yoshida *et al.*, 2002):

We are currently already at 46 references and have not listed this first description of ATE1 in Arabidopsis as we found other articles more relevant to our work. Yoshida *et al.* showed a phenotype of a point mutant in ATE1 in a peculiar genetic accession (Wassilewskija, Ws-0) which per se lacks the second bona fide ATE, that is ATE2, due to a single nucleotide polymorphism causing a premature stop.

(Graciet *et al.*, 2010):

The first description of a molecular function of ATEs is already published Graciet *et al.* PNAS 2009 and we have cited this as reference 34 in line 248. Graciet *et al.* used highly purified extracts of plants of the wild type versus the *ate1 ate2* double mutant in an *in vitro* arginylation assay using an artificial bovine protein substrate lactalbumin. This is all detailed in our text and is the first paper we considered relevant here as we focus on the molecular interplay between and functions of PCOs and ATE1. Graciet *et al.* (2010 plant journal) express Arabidopsis ATEs in an *S. cerevisiae ate1Δ* mutant and show that this is sufficient to destabilize a reporter protein (Asp- or Glu-βGal). Therefore, presence of ATEs lead to destabilization but the mechanism remained unclear. We can add this reference if required but did not find it important in this context.

(Manahan & App, 1973):

This work describes an activity of aminoacyl transfer in rice and wheat extracts. Also arginine gets metabolized but nature of the enzyme and acceptor position remain unclear although it is speculated that the N-terminus could be the acceptor (compare Soffer, JBC 1971). We did not consider this reference as important here in the light of focus and restrictions in reference number.

(Wadas *et al.*, 2016):

This paper was not yet available when we submitted our paper. We have included this reference now.

We have now included all these references in the manuscript together with a careful description of their results and experimental conditions.

In addition the authors cite that the first non Cys-branch Ate1 substrates were only recently identified in mammals (Cha-Molstad 2015, Nat Cell Biol). Prior to this paper has been a number of publications by A. Kashina's group on arginylation targets and the reports on the Ate1 dependent degradation of the caspase generated fragment of BRCA1 by two independent groups (Piatkov et al 2012 PNAS; Xu et al 2012 JBC)

First of all, we thank the Reviewer for highlighting the relevance of Piatkov *et al.* (Piatkov *et al.*, 2012) which we omitted. This paper clearly needs to be mentioned in our work as it molecularly documents and also mechanistically describes the function of N-terminal arginylation on a target protein, BRCA1. The same target was discussed in Xu *et al.* (Xu *et al.*, 2012), without actually demonstrating arginylation but a dependence on presence of ATEs.

The point we are making is that there is a difference between “Ate1-dependent degradation” and “N-terminal arginylation” which also needs to be approached by different experiments. This includes but is not restricted to the different levels of required analytical effort between these two potential outcomes.

The papers from Anna Kashina’s group indeed contain evidences for arginylation but mostly on internal (midchain) arginylation which is not the focus of our work and also not leading to N-end rule degradation. In a recent review (Kashina, 2015), Anna Kashina wrote the following clarifying this:

“[...] ornithine decarboxylase [24–26], BSA, alpha-lactalbumin, and thyroglobulin [14, 27] can be arginylated, [...]”

Even more clearly she stated:

“To date, only one class of naturally occurring targets of this pathway [the authors: the N-end rule pathway] has been proven to exist: several members of the regulators of G-protein signaling (RGS) family [35, 36] [the authors: 35: (Davydov & Varshavsky, 2000); 36: (Lee *et al.*, 2005)], containing Cys in the second position of their coding sequence. It is believed that such RGS proteins undergo removal of N-terminal Met, following by Cys oxidation and arginylation that leads to their decreased metabolic stability. It has been proposed that such Cys-oxidation-dependent degradation participates in oxygen sensing and protection of cells from oxidative stress and that this mechanism may ultimately underlie many RGS-dependent regulatory processes, but overall this mechanism has not been widely studied. ***At the same time, no other in vivo arginylation targets that follow the N-end rule and massively degrade after arginylation have been identified.***”

In the same article, she states the following which is highly relevant to our work:

“N-terminal Cys can also become arginylated and subsequently destabilized and that in the case of Cys, this likely occurs after its in vivo oxidation to ***cysteic acid***, [...]”

Therefore, we think that we deliver important results clarifying both the oxidation state of N-terminal Cys in order to serve as substrate of ATEs. We have concluded our key findings already in the abstract: “This is the first molecular evidence showing N-terminal cysteine oxidation and arginylation by N-end rule pathway components, and the first ATE1 substrate in plants.”

In summary:

Efforts to link the chemical/genetic modulations of PCOs and ATE activity to novel physiological responses and the full biochemical characterization of molecular basis of ATE1/2-mediated arginylation of plant N-end rule substrates in the light of aforementioned comments would improve the scope of current work.

Response: We are grateful for the comments which helped to improve the manuscript and clarify many aspects by further going into detail.

Bibliography

- Bohley P, Kopitz J, Adam G. 1988.** Surface hydrophobicity, arginylation and degradation of cytosol proteins from rat hepatocytes. *Biol Chem Hoppe Seyler* **369 Suppl**: 307-310.
- Ciechanover A, Ferber S, Ganoth D, Elias S, Hershko A, Arfin S. 1988.** Purification and characterization of arginyl-tRNA-protein transferase from rabbit reticulocytes. Its involvement in post-translational modification and degradation of acidic NH₂ termini substrates of the ubiquitin pathway. *J Biol Chem* **263**(23): 11155-11167.
- Davydov I, Varshavsky A. 2000.** RGS4 is arginylated and degraded by the N-end rule pathway in vitro. *J Biol Chem* **275**(30): 22931-22941.
- Dorey C, Cooper C, Dickson DP, Gibson JF, Simpson RJ, Peters TJ. 1993.** Iron speciation at physiological pH in media containing ascorbate and oxygen. *Br J Nutr* **70**(1): 157-169.
- Garman EF, Grime GW. 2005.** Elemental analysis of proteins by microPIXE. *Prog Biophys Mol Biol* **89**(2): 173-205.
- Garzón M, Eifler K, Faust A, Scheel H, Hofmann K, Koncz C, Yephremov A, Bachmair A. 2007.** PRT6/At5g02310 encodes an Arabidopsis ubiquitin ligase of the N-end rule pathway with arginine specificity and is not the CER3 locus. *FEBS Lett* **581**(17): 3189-3196.
- Gibbs DJ, Bacardit J, Bachmair A, Holdsworth MJ. 2014.** The eukaryotic N-end rule pathway: conserved mechanisms and diverse functions. *Trends Cell Biol* **24**(10): 603-611.
- Gibbs DJ, Lee SC, Isa NM, Gramuglia S, Fukao T, Bassel GW, Correia CS, Corbineau F, Theodoulou FL, Bailey-Serres J, et al. 2011.** Homeostatic response to hypoxia is regulated by the N-end rule pathway in plants. *Nature* **479**(7373): 415-418.
- Graciet E, Mesiti F, Wellmer F. 2010.** Structure and evolutionary conservation of the plant N-end rule pathway. *Plant J* **61**(5): 741-751.
- Graciet E, Walter F, Ó'Maoiléidigh D, Pollmann S, Meyerowitz E, Varshavsky A, Wellmer F. 2009.** The N-end rule pathway controls multiple functions during Arabidopsis shoot and leaf development. *Proc Natl Acad Sci U S A* **106**(32): 13618-13623.
- Hamed MY, Keypour H, Silver J, Wilson MT. 1988.** Studies of the Reactions of Iron(II) Ascorbate Mixtures with Molecular-Oxygen in Solution. *Inorganica Chimica Acta-Bioinorganic Chemistry* **152**(4): 227-231.
- Hoernstein SN, Mueller SJ, Fiedler K, Schuelke M, Vanselow JT, Schuessele C, Lang D, Nitschke R, Igloi GL, Schlosser A, et al. 2016.** Identification of targets and interaction partners of arginyl-tRNA protein transferase in the moss *Physcomitrella patens*. *Mol Cell Proteomics*.
- Hu R, Brower C, Wang H, Davydov I, Sheng J, Zhou J, Kwon Y, Varshavsky A. 2006.** Arginyltransferase, its specificity, putative substrates, bidirectional promoter, and splicing-derived isoforms. *J Biol Chem* **281**(43): 32559-32573.
- Hu R, Sheng J, Qi X, Xu Z, Takahashi T, Varshavsky A. 2005.** The N-end rule pathway as a nitric oxide sensor controlling the levels of multiple regulators. *Nature* **437**(7061): 981-986.
- Hu RG, Sheng J, Qi X, Xu Z, Takahashi TT, Varshavsky A. 2005.** The N-end rule pathway as a nitric oxide sensor controlling the levels of multiple regulators. *Nature* **437**(7061): 981-986.
- Imsand EM, Njeri CW, Ellis HR. 2012.** Addition of an external electron donor to in vitro assays of cysteine dioxygenase precludes the need for exogenous iron. *Arch Biochem Biophys* **521**(1-2): 10-17.
- Kaji H, Novelli GD, Kaji A. 1963.** A Soluble Amino Acid-Incorporating System from Rat Liver. *Biochim Biophys Acta* **76**: 474-477.
- Kashina AS. 2015.** Protein Arginylation: Over 50 Years of Discovery. *Methods Mol Biol* **1337**: 1-11.
- Kwon Y, Kashina A, Davydov I, Hu R, An J, Seo J, Du F, Varshavsky A. 2002.** An essential role of N-terminal arginylation in cardiovascular development. *Science* **297**(5578): 96-99.
- Kwon YT, Kashina AS, Davydov IV, Hu RG, An JY, Seo JW, Du F, Varshavsky A. 2002.** An essential role of N-terminal arginylation in cardiovascular development. *Science* **297**(5578): 96-99.
- Lee M, Tasaki T, Moroi K, An J, Kimura S, Davydov I, Kwon Y. 2005.** RGS4 and RGS5 are in vivo

- substrates of the N-end rule pathway. *Proc Natl Acad Sci U S A* **102**(42): 15030-15035.
- Licausi F, Kosmacz M, Weits DA, Giuntoli B, Giorgi FM, Voeselek LA, Perata P, van Dongen JT. 2011.** Oxygen sensing in plants is mediated by an N-end rule pathway for protein destabilization. *Nature* **479**(7373): 419-422.
- Manahan CO, App AA. 1973.** An arginyl-transfer ribonucleic Acid protein transferase from cereal embryos. *Plant Physiol* **52**(1): 13-16.
- McNeill LA, Flashman E, Buck MR, Hewitson KS, Clifton IJ, Jeschke G, Claridge TD, Ehrismann D, Oldham NJ, Schofield CJ. 2005.** Hypoxia-inducible factor prolyl hydroxylase 2 has a high affinity for ferrous iron and 2-oxoglutarate. *Mol Biosyst* **1**(4): 321-324.
- Piatkov K, Brower C, Varshavsky A. 2012.** The N-end rule pathway counteracts cell death by destroying proapoptotic protein fragments. *Proc Natl Acad Sci U S A* **109**(27): E1839-1847.
- Tasaki T, Sriram S, Park K, Kwon Y. 2012.** The N-end rule pathway. *Annu Rev Biochem* **81**: 261-289.
- Tasaki T, Sriram SM, Park KS, Kwon YT. 2012.** The N-end rule pathway. *Annu Rev Biochem* **81**: 261-289.
- Varshavsky A. 2011.** The N-end rule pathway and regulation by proteolysis. *Protein Sci* **20**(8): 1298-1345.
- Wadas B, Piatkov KI, Brower CS, Varshavsky A. 2016.** Analyzing N-terminal Arginylation through the Use of Peptide Arrays and Degradation Assays. *J Biol Chem* **291**(40): 20976-20992.
- Weits D, Giuntoli B, Kosmacz M, Parlanti S, Hubberten H, Riegler H, Hoefgen R, Perata P, van Dongen J, Licausi F. 2014.** Plant cysteine oxidases control the oxygen-dependent branch of the N-end-rule pathway. *Nat Commun* **5**: 3425.
- Xu Z, Payoe R, Fahlman RP. 2012.** The C-terminal proteolytic fragment of the breast cancer susceptibility type 1 protein (BRCA1) is degraded by the N-end rule pathway. *J Biol Chem* **287**(10): 7495-7502.
- Yoshida S, Ito M, Callis J, Nishida I, Watanabe A. 2002.** A delayed leaf senescence mutant is defective in arginyl-tRNA:protein arginyltransferase, a component of the N-end rule pathway in Arabidopsis. *Plant J* **32**(1): 129-137.

REVIEWERS' COMMENTS:

Reviewer #1 (Remarks to the Author):

I am satisfied with the revised manuscript

Reviewer #3 (Remarks to the Author):

After carefully considering the response to previous reviews and the new version of this manuscript, I believe the authors have the authors' have done an thorough job in clarifying the scope of this re-submitted manuscript and have sufficiently addressed my previous concerns and comments.

I have no additional comments regarding this manuscript.

Reviewer 1

I am satisfied with the revised manuscript.

Response: We thank Reviewer 1 for her/his positive comment and the effort and time spent for refereeing.

Reviewer 3

After carefully considering the response to previous reviews and the new version of this manuscript, I believe the authors have the authors' have done an thorough job in clarifying the scope of this re-submitted manuscript and have sufficiently addressed my previous concerns and comments.

I have no additional comments regarding this manuscript.

Response: We thank Reviewer 3 for her/his effort in assessing our manuscript and the comments made. We are happy that the manuscript has improved now and was found to be acceptable.